artificial intelligence/statistics/psychology

peer review, manuscripts, reviewers, statistics, academic journals

# Does peer review improve the statistical content of manuscripts? A study on 27 467 submissions to four journals

Daniel Garcia-Costa[1], Anabel Forte[2], Emilia Lòpez-Iñesta[3], Flaminio Squazzoni[4] and Francisco Grimaldo[1]

[1]Department of Computer Science, and [2]Department of Statistics and Operational Research, University of Valencia, Burjassot, Spain
[3]Department of Mathematics Education, University of Valencia, Valencia, Spain
[4]Department of Social and Political Sciences, University of Milan, Milan, Italy

FS, 0000-0002-6503-6077

Improving the methodological rigour and the quality of data analysis in manuscripts submitted to journals is key to ensure the validity of scientific claims. However, there is scant knowledge of how manuscripts change throughout the review process in academic journals. Here, we examined 27 467 manuscripts submitted to four journals from the Royal Society (2006–2017) and analysed the effect of peer review on the amount of statistical content of manuscripts, i.e. one of the most important aspects to assess the methodological rigour of manuscripts. We found that manuscripts with both initial low or high levels of statistical content increased their statistical content during peer review. The availability of guidelines on statistics in the review forms of journals was associated with an initial similarity of statistical content of manuscripts but did not have any relevant implications on manuscript change during peer review. We found that when reports were more concentrated on statistical content, there was a higher probability that these manuscripts were eventually rejected by editors.

**Author for correspondence:**
Francisco Grimaldo
e-mail: francisco.grimaldo@uv.es

# 1. Introduction

Peer review is key for public trust in the scientific community [1]. By exposing manuscripts to scrutiny by independent experts, it

ensures that scientific claims are grounded on reliable evidence. This requires reviewers to screen the rigour and quality of methods and analysis reported in manuscripts submitted to journals for publication. Although reviewers are expected to check various aspects of a manuscript, this attention to rigour and methodology includes one of the most important imperatives of science as an institutional system— what the famous sociologist of science Robert K. Merton called 'organized skepticism' [2]. While the purposes and practices of peer review have varied considerably with time, place and discipline [3,4], collaboration between unrelated experts in improving the rigour and reliability of scientific findings is of paramount importance especially in the current climate of academic hyper-competition, where scientists are exposed to perverse incentives that maximize the 'publishability' of research rather than its methodological rigour [5–7].

While author–reviewer collaboration during peer review can have different forms, some of which are potentially dysfunctional, e.g. collusion and parochialism [8,9], one of the most important functions of reviewers is to ensure that journals achieve the highest methodological rigour and statistical standards by improving manuscripts. On the one hand, this developmental function of peer review is pivotal in helping authors improve their manuscripts throughout the process [10]. On the other, it enhances the legitimacy and credibility of journals as gatekeepers of scholarly communication [11,12].

Unfortunately, there is little understanding of how this developmental function actually works [13–16]. While research on specific journals has shown that exposure to different rounds of peer review could increase the quality of manuscripts—including later submissions to other journals if rejected [17], other studies have suggested that reviewers are keen to preferably concentrate on theoretical aspects rather than rigour, methodology and statistical content [11,18]. While reviewers are expected to comment on various aspects as well as assisting editors in judging about the suitability of work for publication, exclusively considering background theory, novelty and implications could be detrimental for peer review quality, as reported in the current debate on the quality of peer review during the COVID-19 pandemic [19].

To ensure that reviewers do not only consider novelty as opposed to rigour, journals have introduced guidelines and instructions to ensure they focus on data analysis and statistical testing [14,20]. These often include instructions on how reviewers should provide valid assessments of methods and statistics reported in articles, including measurement validity, outcome sensitivity and findings replicability [21]. While assessing the effective use of these instructions is difficult [22,23], measuring the effect of peer review on how manuscripts change from initial submission to the published version is even more challenging given the system's confidentiality and lack of data on internal editorial processes [24].

To fill this gap, we established a confidential agreement with the Royal Society to access manuscript and peer review data from their journals. The world's oldest independent scientific academy, with the first publication of *Philosophical Transactions* in 1665, the Royal Society pioneered the concepts and practices of academic journals, editorial responsibility and peer review [25]. The Royal Society journals include prestigious titles, such as *Philosophical Transactions A* and *Proceedings A*, which publish research on physical, mathematical and engineering sciences, *Philosophical Transactions B*, *Proceedings B* and *Biology Letters*, with a readership in biological sciences, as well as cross-disciplinary outlets, such as *Interface*, for cross-disciplinary research at the interface between the physical and life sciences, and *Royal Society Open Science*, the Royal Society's most recent open access journal in science, engineering and mathematics.

Data included complete manuscript files and (when available) peer review reports over the same time frame (2006–2017) from all these journals. However, after careful analysis of the database, we restricted our sample to four journals to ensure full comparability of manuscripts (see detail in the Methods section). We concentrated on 27 467 manuscripts from four journals and built a glossary of statistical terms to analyse the text of manuscripts and review reports. Note that in compliance with the agreement signed by all authors of this study, journals were fully anonymized to avoid identification. While other research has examined review reports, e.g. studying their linguistic properties [26–28], our rich and original dataset allowed us to link manuscripts and reports, thus providing a more comprehensive, contextual picture of the collaboration between authors and reviewers in improving manuscripts. Our aim here was to measure the change of the statistical content of manuscripts during peer review, i.e. one of the most relevant functions of reviewers (at least in hard sciences), to estimate conditions and contexts that could stimulate collaborative improvement of manuscripts between authors and reviewers. We first measured the statistical content of manuscripts by scanning their text with a Linguistic Inquiry and Word Count style dictionary built upon a well-known statistics glossary. We assumed that the number of statistical terms included in

**Table 1.** Data overview.

| journal ID | J1 | J7 | J8 | J11 | all |
|---|---|---|---|---|---|
| guidelines for statistics | yes | yes | yes | no | — |
| peer-reviewed manuscripts | 7742 | 350 | 2420 | 731 | 11 243 |
| rejection rate | 59.2% | 47.1% | 57.9% | 49.8% | 58.0% |
| median number of rounds | 1 | 2 | 2 | 2 | 1 |
| mean number of statistical terms | 12.65 | 12.41 | 7.95 | 11.14 | 11.53 |
| desk-rejection or acceptance | 8627 | 957 | 2481 | 1551 | 13 616 |
| mean number of statistical terms | 11.80 | 7.61 | 7.40 | 10.11 | 10.51 |
| manuscripts with no review report | 963 | 429 | 626 | 590 | 2608 |
| rejection rate | 25.5% | 15.9% | 17.4% | 14.2% | 19.4% |
| median number of rounds | 2 | 2 | 3 | 3 | 3 |
| mean number of statistical terms | 12.79 | 11.28 | 8.33 | 10.51 | 10.95 |
| number of research manuscripts | 17 332 | 1736 | 5527 | 2872 | 27 467 |

the text was a proxy of their statistical content. We then measured the statistical content of manuscripts from their initial submissions to their revisions by comparing different versions of the same manuscripts. We also similarly measured the statistical content of review reports. By controlling for important factors, such as the reviewer score received by manuscripts, the number of rounds of peer review and the number of reviewers commenting on the same manuscript, we tried to estimate the effect of peer review on manuscript change and examine the most relevant peer review-related factors shaping the final editorial decision.

Note that we did not assume that any change of the statistical content of manuscripts during peer review would always lead to manuscript improvements in terms of methodological rigour. We also did not assume that any change of statistical terms in the text would necessarily mean the improvement of the quality and rigour of manuscripts. Here, we assumed that the change of statistical content of manuscripts throughout the peer review process as proxied by text revisions may reveal a joint attention effort by reviewers and authors on the methodological content of manuscripts, which is one of the most important functions of peer review. As suggested by recent research, exploring the text of manuscript and peer review reports quantitatively is key to understand the scholarly communication landscape and reconstruct the complex, indirect, collaborative relationship between authors and reviewers, which typically occurs behind the confidentiality of the journal editorial process [29].

## 2. Methods

### 2.1. Data

Data were obtained thanks to a confidential agreement with the Royal Society and were extracted in a comparable time-frame (2006–2017). The original dataset included 60 240 manuscripts submitted to 13 journals. However, in order to ensure full comparability, we concentrated on four journals, which ensured similar standards in terms of number, type of submissions and rejection rates. We also excluded from the sample any manuscript without a clear submission date, being reviewed by multiple journals, changing its status during re-submission, being assigned an unclear final decision in the manuscript submission system (e.g. rejected after accepted or accepted twice), or with missing files. This implied removing more than 24 000 manuscripts from the sample. The remaining 34 781 manuscripts (see table S1 in the electronic supplementary material) were further filtered by selecting all research articles and excluding review papers, opinion pieces, reports, memoirs, recollections, replies etc. We restricted our analysis to journals J1, J7, J8 and J11, since these journals contributed to 97.1% of peer-reviewed manuscripts in our dataset (see electronic supplementary material, table S2). We excluded

**Table 2.** Selected statistical terms for each category.

| category | list of terms |
| --- | --- |
| descriptive | binomial distribution, box plot, density, geometric distribution, histogram, negative-binomial distribution, normal distribution, outlier, percentile, Poisson distribution, quantile, quartile |
| contrast | alternative hypothesis, anova, chi-square, control group, Fisher, multiplicity, null hypothesis, odds, *p*-value, power, rejection region, significant, size effect, *t*-test, *z*-score, *z*-test |
| estimation | average, bias, confidence interval, correlation, estimate, estimation, estimator, expectation, expected value, probability, standard deviation, standard error |
| modelization | area under the curve, association, causality, confounding, cross-sectional study, extrapolation, interaction, interpolation, Kaplan Meier, longitudinal, model, regression |
| generics | Bayes, boostrap, central limit theorem, confidence level, independence, kernel, law of large numbers, likelihood, parameters, population, random, sample, variable |

data from the rest of the journals since they marginally contribute with less than 1% of peer-reviewed articles.

This led us to consider 27 467 manuscripts (table 1), including:

— 11 243 manuscripts that were peer-reviewed,
— 13 616 manuscripts that were desk-rejected or accepted without any round of peer review, and
— 2608 manuscripts without any available review report (i.e. missing or not recorded in the journal submission system).

We then checked whether journals included any guidelines for assessing statistics in their forms sent to reviewers, i.e. an explicit question asking reviewers to assess the quality of a manuscript's statistical analysis in the review form.

To map the statistical content of manuscripts, we selected a list of commonly used statistical terms from a statistics glossary developed by the University of Berkeley (https://www.stat.berkeley.edu/~stark/ SticiGui/Text/gloss.htm). Table 2 shows our selected list of terms, which were then aggregated into five categories for the sake of simplicity. We checked for between-terms orthogonality over the full list of terms, thereby ensuring that each term represented different, not overlapping concepts. Electronic supplementary material, figure S10, shows that between-terms mutual overlapping was rare, except for the term 'model', which has multiple meanings and so was kept in the dictionary.

We applied our dictionary to map the presence of these concepts in the text of manuscripts and review reports by using an R library called `quanteda.dictionaries`. Our study considered all categories together since our main focus was the whole statistical content, regardless of the changing nature of statistical concepts within manuscripts (either descriptive, inferential or both).

We considered the presence of statistical terms within the text of manuscripts and excluded equations, tables and figures, while keeping their captions and recurrences in the text. This allowed us to consider also equations, tables or figures while achieving full comparison of manuscripts and journals and minimizing bias due to either journal- or manuscript-specific features (e.g. different file format, such as PDF, LaTeX, Word, RTF).

## 2.2. Statistical models

To explore the potential effect of peer review on the statistical content of manuscripts and on editorial decisions, we built two models: a Poisson regression for the number of different statistical terms in the

final version of each manuscript which underwent revisions during peer review, and a logistic linear regression for the probability of editorial acceptance of manuscripts after peer review.

We applied a Bayesian variable selection to identify the variables to be included in these linear predictors. To do so, we considered posterior probabilities for each possible combination of variables shown in electronic supplementary material, tables S3 and S4. More specifically, we considered $2^p$ models, $p$ being the potential co-variates in each linear predictor (6 and 7, respectively). We then calculated the posterior inclusion probability (PIP) for each variable as the sum of the posterior probabilities in all models.

This required us to specify the prior distributions involved in the Bayes theorem, which were priors for each model and their parameters, and calculate $2^p$ posterior probabilities which usually need numerical integration (e.g. [30]). Due to the generalized linear nature of our models, we followed [31] and their numerical approximation to the solution, which was implemented in the R package BAS (Bayesian model averaging using Bayesian adaptive sampling) [32].

Tables S3 and S4 in the electronic supplementary material show the results of our model implementations. We selected variables with PIPs greater than 0.5. For the first model, the statistical content of the final version of manuscripts was mainly associated with: the statistical content of the review reports received by manuscripts (max_stats_rev), the level of statistical content in the initial version of manuscripts (initial_stats) and the total number of review rounds undergone by manuscripts (*nrounds*). For the second model, the probability of a manuscript's acceptance was associated with: the statistical content of the associated review reports (max_stats_rev), the number of rounds (nrounds), the number of reviewers (nreviewers) and the review score of manuscripts (score) as defined in [33].

After selecting our model variables, in order to estimate the final number of different statistical terms of manuscripts, we added a random effect per journal to reflect possible differences between journals (figure 1c). However, for the sake of clarity, we excluded this effect when examining the probability of each manuscript's acceptance as these probabilities were similar across journals.

Considering all aspects, the final model of the number of different statistical terms in the final version of manuscripts (*i*) was as follows:

$$y_i \sim \mathrm{Poiss}(\lambda_i)$$
$$\log(\lambda_i) = \beta_0 + \beta_1 \mathrm{max\_stats\_rev}_i + \beta_2 \mathrm{\ initial\_stats}_i + \beta_3 \mathrm{\ nrounds}_i + b_{\mathrm{journal}_i}$$
$$b_j \sim N(0, \sigma) \quad \text{for } j = 1, 7, 8, 11.$$

The selected model for the probability of a manuscript's acceptance (*i*) was as follows:

$$\mathrm{accept}_i \sim \mathrm{Bernoulli}(\pi_i)$$
$$\mathrm{logit}(\pi_i) = \beta_0 + \beta_1 \mathrm{\ max\_stats\_rev}_i + \beta_2 \mathrm{\ nrounds}_i + \beta_3 \mathrm{\ nreviewers}_i + \beta_4 \mathrm{\ score}_i.$$

Following the Bayesian paradigm, all model parameters were considered as random variables and assigned a prior distribution. For the regression coefficients $\beta_j$, we used a normal prior distribution at 0 and with large variance. For the standard deviation of the random effect associated with each journal, $\sigma$, we used a uniform distribution from 0 to 10.

These models were estimated using Bayesian inference through the software JAGS (Just another Gibbs Sampler) and its R interface `rjags` [34]. JAGS performs Markov chain Monte Carlo (MCMC) methods to simulate from desired posterior distributions. After a burning and a thinning MCMC process with one chain, we kept a total of 3000 samples of the posterior distribution of the model parameters.

# 3. Results

Figure 1 shows that initial submissions had a relatively homogeneous statistical content, except for manuscripts directly accepted by editors without any peer review (see the red solid line, which corresponded to 42 manuscripts). The availability of guidelines on statistics for reviewers did not have any qualitative effect on the variation of the initial statistical content of manuscripts submitted for publication (note that journals J1, J7, J8 included these questions in the review form, whereas journal J11 did not). However, we found certain differences between journals, which reflected their different academic audiences. For instance, initial submissions to J7 showed the greatest variability of statistical

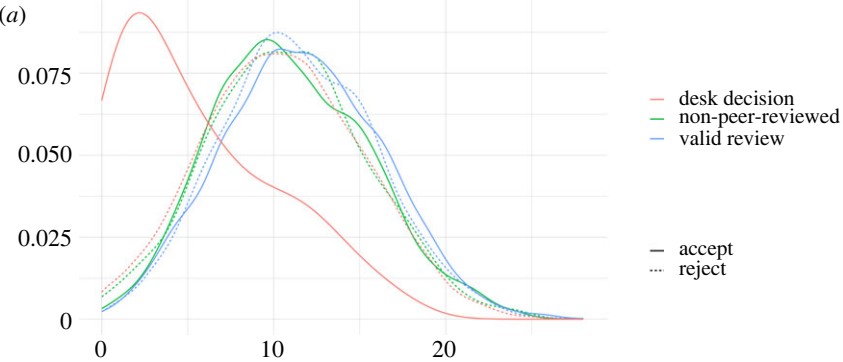

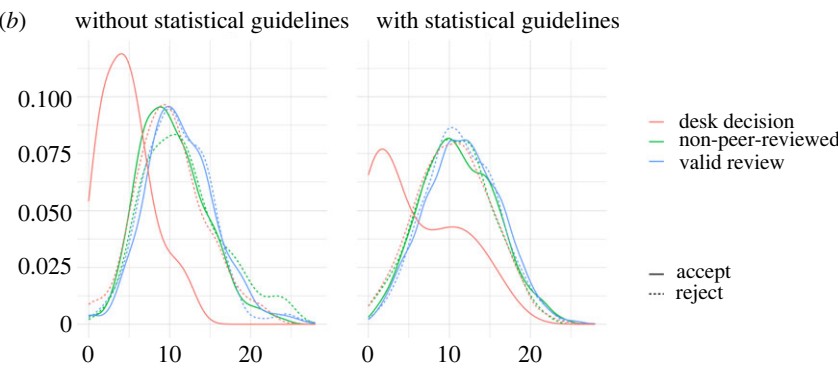

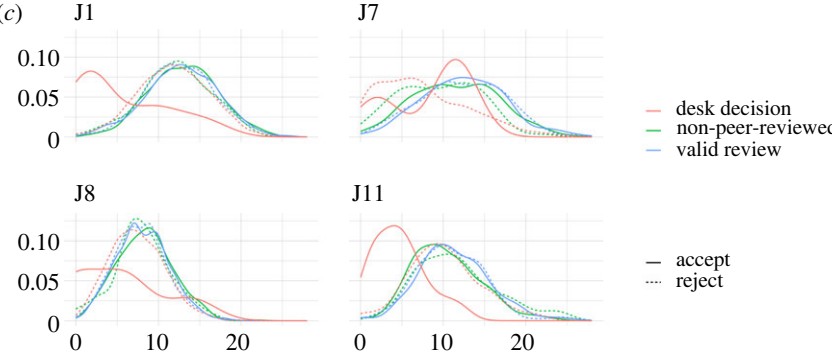

**Figure 1.** Number of different statistical terms (*x*-axis) in initial submissions for rejected (dotted line) or accepted (solid line) manuscripts, in cases of not peer-reviewed (green), desk rejected/accepted (red) and peer-reviewed (blue) manuscripts (*a*), per journal with or without guidelines for statistics (*b*) and per journal (*c*).

content among journals, whereas initial submissions to J8 showed the lowest level of statistical content in the manuscript sample.

We then considered all 11 243 manuscripts that survived the editorial desk and were eventually reviewed multiple times (note that 50.7% of these 11 243 manuscripts were rejected after the first round). We compared their initial statistical content with the final version of manuscripts after peer review. We found that 13.8% of these did not vary their statistical content (i.e. the number of different statistical terms in these manuscripts was the same). For the remaining 35.4%, 23.9% of these manuscripts increased their statistical content, whereas 11.6% reduced it. Regarding the final editorial decision, half of manuscripts accepted for publication increased their statistical content during peer review, 25% decreased it, whereas the remaining 25% did not vary. A proportion of 93.1% of manuscripts which were eventually rejected after peer review did not change in terms of statistical content, 5% increased it, whereas 1.9% decreased it (see figure S1 in the electronic supplementary material).

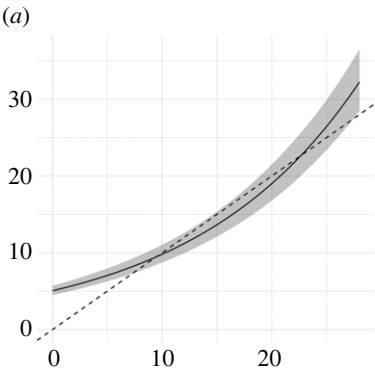 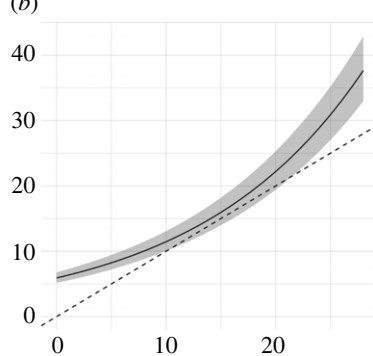

(a) (b)

**Figure 2.** Initial (*x*-axis) versus final (*y*-axis) statistical content of manuscripts by moderate (five terms) statistical content of reports (*a*) or strong (25 terms) statistical content of reports (*b*).

We then considered other variables, which could affect the difference of statistical content during manuscript revisions, including:

— the availability of guidelines to assess the statistical content of manuscripts in the review form of some journals;
— the number of rounds of peer review undergone by manuscripts before the final editorial decision;
— the number of reviewers who jointly or sequentially assessed the same manuscript; and
— the reviewer score, i.e. the quality of manuscripts as assessed by reviewers.

We found that the availability of guidelines in the review form did not have any significant effect on the statistical content of manuscripts (see figure S2 in the electronic supplementary material). We found a positive effect of the peer review on the statistical content of manuscripts: more rounds implied more substantial changes (see figure S3 in the electronic supplementary material). We also found that being assessed by more than two reviewers led to an increase of manuscripts' statistical content (with a significant $\chi^2$-test) for both accepted and rejected manuscripts (see figure S4 in the electronic supplementary material).

We then measured each reviewer's focus on statistics by analysing the statistical content of their comments to authors. Given that this required the availability of review text, we had to restrict our analysis to 11 050 manuscripts (out of 11 243). Results showed that reviewers varied their opinion on the statistical content of manuscripts (see figure S5 in the electronic supplementary material). We found a wide variability in the maximum number of different statistical terms in reviewer reports. Reports with less statistical content were associated with smaller changes in the statistical content of manuscripts (e.g. see the lowest median of statistical terms in review reports—*y*-axis—associated with the value 0 in changes in statistical content of manuscripts—*x*-axis—in figure S5 in the electronic supplementary material).

Following [27,33,35], we used the review score as a proxy of the quality of manuscripts, which is typically a robust predictor of editorial decisions (see detail on the review score in the Methods sections of the references cited above). As expected, editorial decisions on manuscripts depended greatly on review scores: manuscripts rejected after peer review had a lower and more variable review score, whereas manuscripts accepted for publication had higher review scores. Results showed that manuscripts eventually accepted for publication but receiving lowest review scores were also those increasing their statistical content the most during peer review (see figure S6 in the electronic supplementary material).

Results of our models showed that the statistical content of a manuscript's final version was related to the level of statistical content of its initial version submitted for publication, the statistical content of review reports and the number of peer review rounds (see table S3 and figure S7 in the electronic supplementary material, where we report posterior distributions of the exponential of the coefficients associated with each variable). Furthermore, when considering random effects at a journal level, results confirmed that manuscripts submitted to journal J8 generally had lower levels of statistical content (see figure S8 in the electronic supplementary material).

More importantly, we found that reviewers contributed to increase the statistical content of manuscripts regardless of the statistical content of review reports (figure 2). However, it is worth noting that manuscripts with moderate levels of initial statistical content (i.e. about 15 words compared to the maximum number of different statistical terms, which was 30 terms as shown in (figure 1) had fewer variations throughout the peer review process than those with either a small or large number of different statistical terms in their initial version. In short, manuscripts with initial low or high levels of statistical content were those which improved the most during peer review.

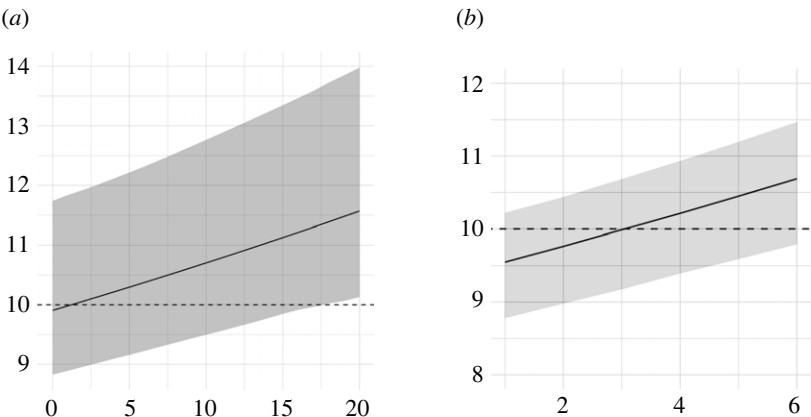

**Figure 3.** Number of different statistical terms in the final version of manuscripts (*y*-axis) as due to (*x*-axis) the maximum number of different statistical content in the report (*a*) and the number of rounds of peer review (*b*).

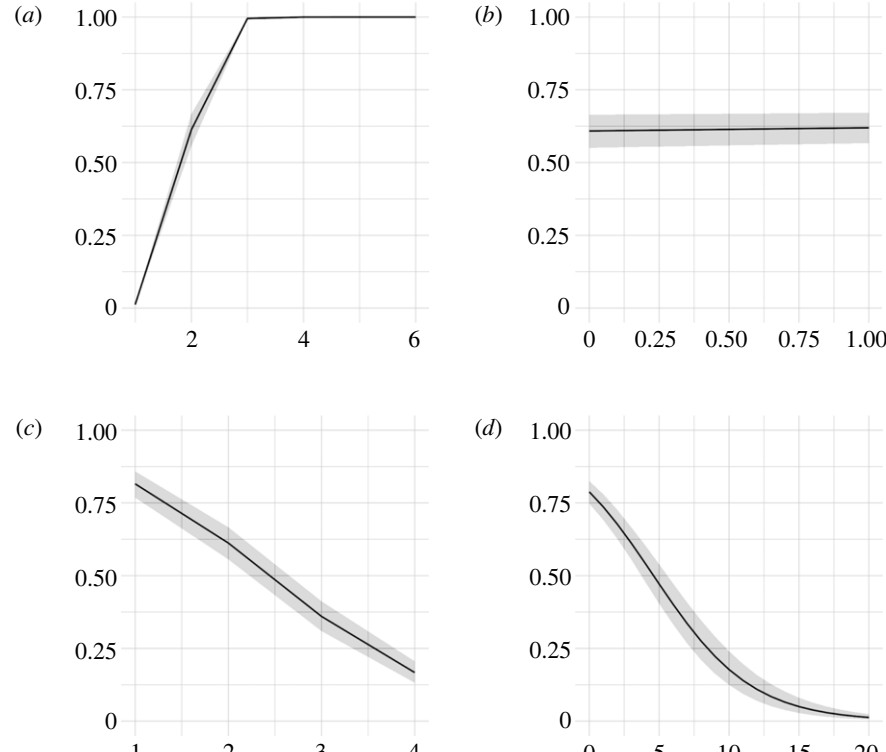

**Figure 4.** The probability of a manuscript's acceptance (*y*-axis) due to the number of peer review rounds (*a*), the review score for papers following two rounds of review (*b*), the number of reviewers (*c*) and the maximum number of different statistical terms in the review reports (*d*).

Figure 3 shows that the effect of the number of different statistical terms in the reports and the number of rounds of peer review on the final statistical content of manuscripts is increasing and linear. Though, changes were mostly marginal, e.g. adding one new term to the average increase of 10 different statistical terms (see dotted line) in the final version of the manuscript.

Electronic supplementary material, table S4, shows the results of our logistic regression model (see electronic supplementary material). The probability of a manuscript being accepted for publication was related to the number of reviewers who assessed it, the statistical content of review reports, the overall opinion of reviewers (i.e. the review score received by the manuscript in all rounds of peer review), and the number of rounds of reviews (see the posterior distributions of the exponential of the coefficients associated with each of the variables in figure S9 in the electronic supplementary material).

Figure 4 shows that a manuscript that underwent more than two review rounds was eventually accepted by the editor (figure 4*a*). The review score was increasingly instrumental for a manuscript's

final acceptance as it is closely related to the number of rounds. For instance, manuscripts undergoing only one round of reviews had a median review score of 0.12 and those undergoing more than one round had a median review score of 0.67. Although the effect of the number of rounds was strongly associated with the review score (e.g. see the marginal effect for manuscripts undergoing two review rounds, figure 4b), we considered both variables to build a better model, as indicated by their posterior inclusion probabilities (see electronic supplementary material, table S4). We also found a decreasing effect of the number of reviewers (figure 4c) and the statistical content of reviews (figure 4d). This would suggest that the more the reviewers were concentrated on statistics in their reports, the less likely a manuscript was eventually accepted for publication by the editor.

# 4. Discussion

The role of peer review in improving the quality of scientific publications has been subject to increasing scrutiny in recent research [28], which has led especially to more examination of current practices and standards [27,36,37]. However, this type of research only rarely integrates full data on manuscripts during each stage of the editorial process and data on review reports, at the same time covering different journals [38,39]. Integrating data on manuscripts and reports is key to providing a context-specific picture of peer review and editorial processes, not to mention the possibility of assessing changes and revisions of manuscripts due to peer review [28,40]. Although difficult, pooling across-journal data is instrumental to examine the emergence of peer review practices that are shared in various communities [24,35].

Here, we aimed to fill this gap by examining manuscript changes and peer review reports in a sample of manuscripts submitted to four journals from the Royal Society in the same time frame (2006–2017). We concentrated on the statistical content of manuscripts as a proxy of the rigour of the analysis supporting scientific claims and findings in published manuscripts. While this can be irrelevant in certain areas of research, e.g. the humanities, robust quantitative methodologies and statistical tests are key to corroborate findings in 'hard sciences'. Furthermore, our database allowed us to consider various factors that could influence manuscript development, including the number of rounds of peer review undergone by manuscripts, the number of reviewers who jointly or sequentially assessed them, the reviewer score, reflecting a manuscript's perceived quality by reviewers, and the availability of guidelines in the reviewer form.

Our results suggest that manuscripts with both initial lowest or highest levels of statistical content increased their statistical content during the process, whereas desk-rejected manuscripts had comparatively fewer statistical terms in their text. We found that these developments were associated with a higher probability of a manuscript's acceptance. The availability of reviewer guidelines on statistics on review forms seems to ensure similar initial levels of statistical content among submitted manuscripts but did not have any qualitative implication on manuscript change during peer review. We found that editors were more likely to reject manuscripts when reviewers concentrated more on the statistical content of manuscripts in their reports.

Note that our developmental measurements of peer review here did not consider the possible developments of manuscripts rejected by these four journals but later submitted to and possibly published by other journals. Although authors can disregard advice from reviewers after rejection and rejections are costly to the system and are often a source of academic frustration [41], research on the fate of rejected manuscripts has found that manuscripts are often developed across journals via subsequent, multiple submissions [17,42]. Review reports are of a great benefit to authors' learning and a source of scientific improvement, especially when reviewers spot flaws in methodology and lack of rigour in analysis, i.e. amendable weaknesses [43].

This said, our study has certain limitations. First, in order to analyse the text of manuscripts and review reports, we started from a glossary of statistical terms, selected those relevant to our purposes and measured the occurrence of these terms throughout manuscripts and reports. In our opinion, this was an appropriate design strategy considering the type of journals and areas of research in our dataset and the fact that statistics is a standardized field. However, integrating our measurements with qualitative analysis of the text by human experts would be a significant step forward [40]. This would also help to assess the potentially negative effect of reviewer requests on manuscript change as well as inform us about the link between increased statistical content and methodological quality and rigour of reported studies. Furthermore, applying supervised machine learning techniques could also be helpful to test alternative measurements. Unfortunately, yet large, and complete, our dataset was

not sufficiently large to use supervised machine learning techniques, e.g., neural networks, which require large-scale, training datasets.

Secondly, although the four journals from the Royal Society covered here allowed us a certain degree of variety in terms of fields and journals, extending our research to other fields where statistics and statistical models are important, such as medicine, engineering, economics and social sciences, could help provide a more comprehensive picture of the developmental function of peer review in terms of rigour and methodology. This would also increase the in-depth definition of rigour: in certain areas, it is expected that the concept of rigour could extend to hypothesis testing and data collection, thereby suggesting that looking at statistical terms is only an approximation.

Finally, note that this type of research on language and content analysis of manuscripts and reports is in its infancy [26,28,44–46]. This implies that any measurement is only explorative and caution must be used when drawing any conclusions from a study's findings. On the one hand, even research on manuscript change in preprint–publication pairs estimates the potential effects of peer review only indirectly as the link between manuscripts and reports is missing [40,47,48]. On the other hand, research on the content of peer review reports from available report repositories, e.g. Publons, cannot help to estimate the effect of reports on manuscript change due to lack contextual information on associated manuscripts [49]. To improve this type of research, removing obstacles against data sharing from publishers to the community and increasing interdisciplinary, multi-approach studies combining qualitative and quantitative research is needed [24]. Not only would this help us assess the developmental role of peer review more systematically, but also this type of research could inform guidelines and arrangements to improve the fairness of peer review [28,29] and improve our understanding of the multiple functions and dimensions of this complex social institution called peer review [4,50].

Data accessibility. The dataset used for this study and the code for replication are available at https://doi.org/10.7910/DVN/MOKJED.

Supplementary material is available online [51].

Authors' contributions. D.G.-C: conceptualization, data curation, formal analysis, methodology, writing—original draft, writing—review and editing; E.L.-I.: conceptualization, formal analysis, writing—original draft, writing—review and editing; A.F.: conceptualization, formal analysis, writing—original draft, writing—review and editing; F.S.: conceptualization, methodology, supervision, writing—original draft, writing—review and editing; F.G.: conceptualization, data curation, project administration, supervision, writing—original draft, writing—review and editing.

All authors gave final approval for publication and agreed to be held accountable for the work performed therein.

Conflict of interest declaration. The authors declare no competing interests.

Funding. This work was partially supported by the Spanish Ministry of Science and Innovation (MCINN), the Spanish State Research Agency (AEI) and the European Regional Development Fund (ERDF) under projects RTI2018-095820-B-I00 and PID2019-104790GB-I00. F.S. was supported by a grant of MIUR-Italian Minister for Education, University and Research (20178TRM3F_002) and a grant from the University of Milan (PSR2015-17 transition grant). Funders had no role in the design of this study.

Acknowledgements. We gratefully acknowledge Phil Hurst and the team of the Royal Society for providing data and covering the cost of their extraction from manuscript submission systems.

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
