## [Peer Review File · Royal Society Open Science]

Review History

RSOS-210681.R0 (Original submission)

Review form: Reviewer 1 (Ludo Waltman)

Is the manuscript scientifically sound in its present form?

Yes

Are the interpretations and conclusions justified by the results?

No

Is the language acceptable?

Yes

Do you have any ethical concerns with this paper?

No

Have you any concerns about statistical analyses in this paper?

No

Recommendation?

Major revision is needed (please make suggestions in comments)

Comments to the Author(s)

This paper studies how peer review affects the number of statistical terms used in articles published in four journals of the Royal Society. The paper is interesting, but several improvements need to be made.

I find the following statement in the introduction quite questionable: “the supreme function of reviewers is to help journals achieve the highest methodological rigour and statistical standards of scientific publications”. Peer review and peer reviewers have many different functions. Improving methodological and statistical quality is one of them, but there are many others as well (e.g., improving the conceptual design of the research under review, improving the interpretation of empirical findings, improving the embedding in the literature, improving the presentation of the research under review, assessing the relevance of the article under review for a particular research community, advising a journal editor on the suitability of the article under review for publication in a journal, etc.).

The authors do not reveal the identity of the four journals included in the analysis (although they do mention that the journals are published by the Royal Society). Different research fields use different scientific methods. Statistical methods are just one type of scientific method. There are many others as well (e.g., machine learning, computer simulation, agent-based modeling, mathematical modeling, qualitative research methods, etc.). Since the identity of the four journals is not revealed, it is unclear to what extent these journals publish research from fields that do indeed rely strongly on statistical methods. It is therefore hard to assess whether the focus of the authors on statistical methods (as opposed to other types of scientific methods) is a sensible choice. Knowing the identity of the four journals would be highly informative for readers of the paper. I therefore hope the authors are able to reveal the identity of the journals. If this is not possible, the authors should at least provide a brief explanation of the research fields on which each of the four journals is focused and of the different scientific methods used in these research fields.

For many of the results discussed in the results section, the actual results are not shown in this section. Instead, they are shown in tables and figures that are made available as supplementary information. This makes the results section hard to read. It would be much easier for the reader if all results, or at least most of them, are shown in the paper itself rather than in the supplementary information.

The figures are difficult to read. The authors should use a much larger font size.

“We concentrated here on ... in published manuscripts”: Generalizing from ‘statistical content’ to ‘quality and rigour of the analysis’ is quite a big step. The authors need to reflect more carefully on this. To what extent does ‘statistical content’ indeed offer a reasonable proxy of ‘quality and rigour of the analysis’? I believe that statistical content is an important element of quality and rigour (at least in some research fields), but various other elements are important as well (e.g., design of the analysis, data collection approach, interpretation of the results, etc.), and these elements are not covered by the analysis of the authors.

“the most common terms in all statistical methodologies”: This is too strong and needs to be rephrased (e.g., “commonly used statistical terms”). Various statistical methodologies are not covered by the terms in Table 2, for instance commonly used multivariate analysis methods such as multidimensional scaling, principal component analysis, factor analysis, correspondence analysis, etc., statistical learning methods such as k-means clustering, hierarchical clustering, discriminant analysis, etc., and probably many other methodologies as well.

“To understand which variables needed to be included”: Please list the variables that are taken into consideration.

Review form: Reviewer 2

Is the manuscript scientifically sound in its present form?

No

Are the interpretations and conclusions justified by the results?

No

Is the language acceptable?

Yes

Do you have any ethical concerns with this paper?

No

Have you any concerns about statistical analyses in this paper?

Yes

Recommendation?

Major revision is needed (please make suggestions in comments)

Comments to the Author(s)

The confidentiality of peer review is one of the major challenges to scholarly understanding of its operations, fairness and effectiveness. This paper is able to make use of a fantastic dataset: access to original submitted papers, final versions AND the peer review reports is very rare, and it has the potential to enable fruitful investigation of peer review in action. The way in which peer review changes the statistical content of papers is just one of the many aspects that could be investigated.

This paper seeks to investigate how reviewers' attention to the statistical content of submitted papers correlates to subsequent revisions made by authors to the statistical content of their papers. It looks also at how the number of rounds of review, and the final acceptance or rejection decision, affects changes to the statistical content. This is potentially interesting, but this reader needs more narrative analysis to draw out the implications of the data presented here, and what human practices (of peer review) the authors believe are implied by the data.

My own expertise lies in the humanistic and social sciences, not in computer science nor statistics. I can see that there has been substantial work done to calculate the number of statistical terms used in tens of thousands of papers and reviews, to calculate their variation, and the changes made during the review process. But what does it all mean? Is this statistical analysis answering an interesting or relevant question?

For this paper to be interesting to the wider community of those studying peer review, it would need more discussion/explanation of the measure used, what it might actually mean, and what this tells us about the role of statistics in scientific papers in the mind of authors and reviewers. At present, the 'discussion' (section 3) doesn't add very much to the description of results (section 2).

Among the questions that I had were:

- Why is 'number of statistical terms mentioned' assumed to be a meaningful measure of statistical content? [for instance, why is it more meaningful than 'proportion of the article devoted to graphs and/or paragraphs of text with statistical terms in them'?).
- What is the justification for assuming that 'more statistical terms mentioned' = 'better', and 'fewer terms' = 'weak' [as for J8]? This seems like a tacit assumption coming from authors based in a statistics-heavy discipline. For instance, there are clear disciplinary variations in the amount of statistical content that is required/expected, and the Royal Society does publish journals with content from a wide variety of disciplines. So, what makes the statistical content of J8 'weak' rather than 'of a different discipline'? And would a very long paragraph full of statistical terms that was, nonetheless, complete rubbish still count as 'better' because it has more hits for the key terms?
- Did submitted papers with lots of statistical content generate reviewer reports with lots of commentary on statistics? And did submitted papers with little statistical content generate reviewer reports with lots of comments on statistics (implying: more statistics required) or not (implying: this is normal in this field)?
- How can we know whether reviewer reports with lots of statistical terms in them were critiquing or appreciating the statistics in the submitted paper?
- Did the three journals with guidance to reviewers asking them to look at statistics have that guidance in place throughout the time period studied? [2006 seems quite early to have that in place...] Did the introduction of such guidance lead to any change of behaviour among reviewers? Or among authors making revisions?

Review form: Reviewer 3 (Peiling Wang)

Is the manuscript scientifically sound in its present form?

No

Are the interpretations and conclusions justified by the results?

No

Is the language acceptable?

Yes

Do you have any ethical concerns with this paper?

No

Have you any concerns about statistical analyses in this paper?

Yes

Recommendation?

Major revision is needed (please make suggestions in comments)

Comments to the Author(s)

This is a very interesting project with a unique dataset.

The manuscript falls slightly short; I hope the authors will revise to strengthen such a valuable research output.

The first issue is the structure. I wondered why the authors deviated from the instruction to the author and the journal-provided template?

Currently, the Results and Discussion sections preceded the Methods. The Conclusions section is missing. This unusual order of contents made the manuscript hard to follow. Please use the template at

https://royalsocietypublishing.org/pb-assets/RSOS%20assets/RSOS_template-1580479300553.docx

I suggest providing a conceptual framework to relate specific research questions with the statistical analysis. Please state clearly the hypotheses to connect variables (such as dependent with independent) in order to justify the chosen statistics, if the intention was to test if there was a significant effect on ... (e.g., Page 5, Line 4-5; Page 9, Line 42)

In terms of the first step of data processing the six categories of LIWC, I wondered if the authors did manually or applied a computer program (Page 9, Line 15-). Had the processed data been illustrated, readers could perhaps make sense of the analysis or results. I tried to open the dataset following the link but, the link is inactive in either the html or the pdf version:

B Data availability

The dataset used for this study is available to reviewers here.

I am researching peer review using OPR. Therefore, I have basic ideas of what review reports are. But the authors did not share or illustrate how they coded data using the six categories. Some specifics on methods were missing.

Although valuable work, the authors did not write according to the journal's guidelines, nor following statistical convention: Research questions  hypothesis 

Decision letter (RSOS-210681.R0)

Dear Professor Squazzoni,

The Editors assigned to your paper RSOS-210681 "Does peer review improve the statistical content of manuscripts? A study on 27,467 manuscripts" have now received comments from reviewers and would like you to revise the paper in accordance with the reviewer comments and any comments from the Editors. Please note this decision does not guarantee eventual acceptance.

We invite you to respond to the comments supplied below and revise your manuscript. Below the referees' and Editors' comments (where applicable) we provide additional requirements.

Final acceptance of your manuscript is dependent on these requirements being met. We provide guidance below to help you prepare your revision.

Please submit your revised manuscript and required files (see below) no later than 21 days from today's (ie 25-Jun-2021) date. Note: the ScholarOne system will 'lock' if submission of the revision is attempted 21 or more days after the deadline. If you do not think you will be able to meet this deadline please contact the editorial office immediately.

on behalf of Professor Mark Girolami (Associate Editor) and Marta Kwiatkowska (Subject Editor)
openscience@royalsociety.org

Associate Editor Comments to Author (Professor Mark Girolami):

Three experts have provided reports on this submission and all are agreed that substantial revisions are required before it can be considered further. Please respond to and address in detail all of the comments and suggestions provided by the referees in a major revision.

Reviewer comments to Author:

Reviewer: 1

Comments to the Author(s)

This paper studies how peer review affects the number of statistical terms used in articles published in four journals of the Royal Society. The paper is interesting, but several improvements need to be made.

I find the following statement in the introduction quite questionable: "the supreme function of reviewers is to help journals achieve the highest methodological rigour and statistical standards of scientific publications". Peer review and peer reviewers have many different functions. Improving methodological and statistical quality is one of them, but there are many others as well (e.g., improving the conceptual design of the research under review, improving the interpretation of empirical findings, improving the embedding in the literature, improving the presentation of the research under review, assessing the relevance of the article under review for a particular research community, advising a journal editor on the suitability of the article under review for publication in a journal, etc.).

The authors do not reveal the identity of the four journals included in the analysis (although they do mention that the journals are published by the Royal Society). Different research fields use different scientific methods. Statistical methods are just one type of scientific method. There are many others as well (e.g., machine learning, computer simulation, agent-based modeling, mathematical modeling, qualitative research methods, etc.). Since the identity of the four journals is not revealed, it is unclear to what extent these journals publish research from fields that do indeed rely strongly on statistical methods. It is therefore hard to assess whether the focus of the authors on statistical methods (as opposed to other types of scientific methods) is a sensible choice. Knowing the identity of the four journals would be highly informative for readers of the paper. I therefore hope the authors are able to reveal the identity of the journals. If this is not possible, the authors should at least provide a brief explanation of the research fields on which each of the four journals is focused and of the different scientific methods used in these research fields.

For many of the results discussed in the results section, the actual results are not shown in this section. Instead, they are shown in tables and figures that are made available as supplementary information. This makes the results section hard to read. It would be much easier for the reader if all results, or at least most of them, are shown in the paper itself rather than in the supplementary information.

The figures are difficult to read. The authors should use a much larger font size.

“We concentrated here on ... in published manuscripts”: Generalizing from ‘statistical content’ to ‘quality and rigour of the analysis’ is quite a big step. The authors need to reflect more carefully on this. To what extent does ‘statistical content’ indeed offer a reasonable proxy of ‘quality and rigour of the analysis’? I believe that statistical content is an important element of quality and rigour (at least in some research fields), but various other elements are important as well (e.g., design of the analysis, data collection approach, interpretation of the results, etc.), and these elements are not covered by the analysis of the authors.

“the most common terms in all statistical methodologies”: This is too strong and needs to be rephrased (e.g., “commonly used statistical terms”). Various statistical methodologies are not covered by the terms in Table 2, for instance commonly used multivariate analysis methods such as multidimensional scaling, principal component analysis, factor analysis, correspondence analysis, etc., statistical learning methods such as k-means clustering, hierarchical clustering, discriminant analysis, etc., and probably many other methodologies as well.

“To understand which variables needed to be included”: Please list the variables that are taken into consideration.

Reviewer: 2

Comments to the Author(s)

The confidentiality of peer review is one of the major challenges to scholarly understanding of its operations, fairness and effectiveness. This paper is able to make use of a fantastic dataset: access to original submitted papers, final versions AND the peer review reports is very rare, and it has the potential to enable fruitful investigation of peer review in action. The way in which peer review changes the statistical content of papers is just one of the many aspects that could be investigated.

This paper seeks to investigate how reviewers’ attention to the statistical content of submitted papers correlates to subsequent revisions made by authors to the statistical content of their papers. It looks also at how the number of rounds of review, and the final acceptance or rejection

decision, affects changes to the statistical content. This is potentially interesting, but this reader needs more narrative analysis to draw out the implications of the data presented here, and what human practices (of peer review) the authors believe are implied by the data.

My own expertise lies in the humanistic and social sciences, not in computer science nor statistics. I can see that there has been substantial work done to calculate the number of statistical terms used in tens of thousands of papers and reviews, to calculate their variation, and the changes made during the review process. But what does it all mean? Is this statistical analysis answering an interesting or relevant question?

For this paper to be interesting to the wider community of those studying peer review, it would need more discussion/explanation of the measure used, what it might actually mean, and what this tells us about the role of statistics in scientific papers in the mind of authors and reviewers. At present, the 'discussion' (section 3) doesn't add very much to the description of results (section 2).

Among the questions that I had were:

- Why is 'number of statistical terms mentioned' assumed to be a meaningful measure of statistical content? [for instance, why is it more meaningful than 'proportion of the article devoted to graphs and/or paragraphs of text with statistical terms in them'?).
- What is the justification for assuming that 'more statistical terms mentioned' = 'better', and 'fewer terms' = 'weak' [as for J8]? This seems like a tacit assumption coming from authors based in a statistics-heavy discipline. For instance, there are clear disciplinary variations in the amount of statistical content that is required/expected, and the Royal Society does publish journals with content from a wide variety of disciplines. So, what makes the statistical content of J8 'weak' rather than 'of a different discipline'? And would a very long paragraph full of statistical terms that was, nonetheless, complete rubbish still count as 'better' because it has more hits for the key terms?
- Did submitted papers with lots of statistical content generate reviewer reports with lots of commentary on statistics? And did submitted papers with little statistical content generate reviewer reports with lots of comments on statistics (implying: more statistics required) or not (implying: this is normal in this field)?
- How can we know whether reviewer reports with lots of statistical terms in them were critiquing or appreciating the statistics in the submitted paper?
- Did the three journals with guidance to reviewers asking them to look at statistics have that guidance in place throughout the time period studied? [2006 seems quite early to have that in place...] Did the introduction of such guidance lead to any change of behaviour among reviewers? Or among authors making revisions?

Reviewer: 3

Comments to the Author(s)

This is a very interesting project with a unique dataset.

The manuscript falls slightly short; I hope the authors will revise to strengthen such a valuable research output.

The first issue is the structure. I wondered why the authors deviated from the instruction to the author and the journal-provided template?

Currently, the Results and Discussion sections preceded the Methods. The Conclusions section is missing. This unusual order of contents made the manuscript hard to follow. Please use the template at

https://royalsocietypublishing.org/pb-assets/RSOS%20assets/RSOS_template-1580479300553.docx

I suggest providing a conceptual framework to relate specific research questions with the statistical analysis. Please state clearly the hypotheses to connect variables (such as dependent with independent) in order to justify the chosen statistics, if the intention was to test if there was a significant effect on ... (e.g., Page 5, Line 4-5; Page 9, Line 42)

In terms of the first step of data processing the six categories of LIWC, I wondered if the authors did manually or applied a computer program (Page 9, Line 15-). Had the processed data been illustrated, readers could perhaps make sense of the analysis or results. I tried to open the dataset following the link but, the link is inactive in either the html or the pdf version:

B Data availability

The dataset used for this study is available to reviewers here.

I am researching peer review using OPR. Therefore, I have basic ideas of what review reports are. But the authors did not share or illustrate how they coded data using the six categories. Some specifics on methods were missing.

Although valuable work, the authors did not write according to the journal's guidelines, nor following statistical convention: Research questions  hypothesis 

===PREPARING YOUR MANUSCRIPT===

===PREPARING YOUR REVISION IN SCHOLARONE===

Author's Response to Decision Letter for (RSOS-210681.R0)

See Appendix A.

RSOS-210681.R1 (Revision)

Review form: Reviewer 1 (Ludo Waltman)

Is the manuscript scientifically sound in its present form?

Yes

Are the interpretations and conclusions justified by the results?

Yes

Is the language acceptable?

Yes

Do you have any ethical concerns with this paper?

No

Have you any concerns about statistical analyses in this paper?

No

Recommendation?

Accept as is

Comments to the Author(s)

I am satisfied with the revised version of the paper. I consider the paper suitable for publication.

It is a pity that the identity of the four journals cannot be revealed. I wonder whether the Royal Society could give permission to the authors to reveal the identity of the journals. This information would substantially enrich the paper, while I believe it wouldn't harm the interests of the journals in any way.

Review form: Reviewer 4

Is the manuscript scientifically sound in its present form?

No

Are the interpretations and conclusions justified by the results?

Yes

Is the language acceptable?

Yes

Do you have any ethical concerns with this paper?

Yes

Have you any concerns about statistical analyses in this paper?

Yes

Recommendation?

Reject

Comments to the Author(s)

The authors have access to an exciting data set, but sadly the quality of the paper does not live up to its promise. The main issue is the one raised previously by reviewer 2. It is not sufficiently clear why the authors choose the particular variables and engage in the particular analyses that they do. The introduction and discussion sections make bold claims about peer review and its purpose, but the connection with the novel content of the paper remains far too tacit. How should we interpret the number of statistical terms used in a given manuscript and how that changes in response to review? What is the connection with improving manuscripts (which the paper makes much of in the introduction and discussion)?

A minor comment: tables S3 and S4 seem to break the author's confidentiality agreement, as it is easy to tell from either of these tables that J1 is by far the largest journal and J8 the second largest. An informed reader could easily work out which Royal Society journals they are. This kind of disaggregated data by journal level should not be provided if the individual journals are meant to be anonymous. I'm not sure how to fix S3, but S4 could be easily fixed by focusing on the percentage of peer-reviewed manuscripts within each journal (i.e., making the columns add to 100% instead of the rows), which seems more relevant and interesting anyway.

Review form: Reviewer 5 (Noah van Dongen)**Is the manuscript scientifically sound in its present form?**

Yes

Are the interpretations and conclusions justified by the results?

No

Is the language acceptable?

Yes

Do you have any ethical concerns with this paper?

No

Have you any concerns about statistical analyses in this paper?

No

Recommendation?

Major revision is needed (please make suggestions in comments)

Comments to the Author(s)

Dear authors,

I read your manuscript with great interest, but I have some concerns. I know that this manuscript has been reviewed before and if you already addressed my comments to the satisfaction of other reviewers please let me know (plus tell me how you addressed them, please). As I understand it, your aim was to investigate the extent to which peer review affected the manuscripts' methodological rigor. As a proxy for methodological rigor, you used the number of statistical terms in the manuscripts. I have several concerns about the adequacy of this proxy and a number of things I could not glean from the text and would prefer to see clarified. I list these concerns and questions below.

My apologies for not being more positive and I hope my comments help in improving the manuscript.

Sincerely,
Noah van Dongen

Concerns about how you operationalize methodological rigor

Could you please explain why the number of statistical terms in a manuscript is a good indicator of methodological rigor. There might be very good reasons for using this proxy, but I did not see them expressed in the manuscript. I think the readers of your work need to see some kind of clarification on the relationship between methodological rigor and the number of statistical terms in a manuscript.

In general, research has been completed when the manuscript has been submitted for peer review. If this is so, could you please explain how peer review can affect the methodological rigor of the research? As I see it at the moment, and as long as no additional experiments are conducted, peer review can only affect the report quality of the research and the adequacy of the data processing and statistical analyses.

On page 3, you write that you generated six categories "four of them referring to various statistical approaches." However, if I am not mistaken, in the rest of the manuscript no reference is made to these categories. It seems like only the total number of statistical terms are used and not the number of terms per category. If this is correct, I would like to see explained why this is. Also, are all the terms from the six categories used, or just the terms from the four categories referring to the statistical approaches?

Why are the absolute and not the relative numbers of statistical terms used? As I read the manuscript, you use counts of the identified statistical terms and not the number of terms with respect to, for instance, the length of the manuscript. Could you please provide some argumentation for this choice? I do not think you should categorize your manuscripts in 'low', 'moderate', and 'high' statistical content, if manuscripts can be of different lengths or can vary in the number of studies they report. Also, on occasion, you call manuscripts with low statistical content 'manuscripts with weak statistical content.' At the moment, I am missing the justification for using the term 'weak.'

Some missing information I would like to see cleared up

How were the categories of 'low', 'moderate', and 'high' statistical content identified/defined? On page 6 you state that a moderate level of initial statistical content is "around 15 words", but I

did not read anything about how this was defined and why this is a moderate number of statistical terms (or in relation to what is this a moderate number).

When you mention that a certain percentage of manuscripts did not vary in statistical content between reviews, do you mean that the statistical terms are exactly the same, the number of terms is exactly the same, or the number of terms is approximately the same?

On page 7, you write “While these differences were on average minimal, we found cases where the number of statistical terms in the text greatly increased throughout a manuscript’s revision process.” What are you trying to convey in this sentence? Are you saying that there were outliers? I think this sentence needs specific numbers and/or further elaboration.

On page 7, you write about Figure 4 “the review score was increasingly instrumental for a manuscript’s final acceptance, yet the effect was not particularly prominent (top right)” As I read Figure 4, ‘not particularly prominent’ is an understatement. If I read the graph correctly, the probability of acceptance changed on average only a few percent from a reviewer score of zero to max. Is this correct?

The last sentence of the first paragraph of the discussion starts with “on the other hand”. I am sorry, but I do not understand to what this sentence is a contrast.

On page 7, you write “We found that these developments were instrumental to increase the probability of a manuscript’s acceptance.” I think most people would interpret the word ‘instrumental’ as the effect being both causal and strong/significant/extensive. Do you think the use of this term is warranted? Also, as I read it, (the analysis of) this causal connection is not specified in the manuscript. Could this be specified?

On page 7, you write “The availability of reviewer guidelines on statistics on review forms increased the similarity of statistical content among submitted manuscripts but did not have any qualitative effect on manuscript development during peer review.” Do you mean that different manuscripts were similar in their statistical content, or that each of these manuscripts did not change during the review process? Also, I don't see this result in the results section or in Figure 1. Or am I missing something?

Minor comments

In the last equation of page 4: should it not be $P(x_2 | D)$ (instead of X_1)?

On page 7, you write “four journals from The Royal Society over a similar time frame (2006-2017).” Don’t you mean ‘the same time frame’?

You end the manuscript with the limitations. Would it not be better to end with some general conclusion, suggestions for the future, or some kind of take-home-message?

Decision letter (RSOS-210681.R1)

Dear Prof Squazzoni

The Editors assigned to your paper RSOS-210681.R1 "Does peer review improve the statistical content of manuscripts? A study on 27,467 manuscripts" have now received comments from reviewers and would like you to revise the paper in accordance with the reviewer comments and any comments from the Editors. Please note this decision does not guarantee eventual acceptance.

Please submit your revised manuscript and required files (see below) no later than 21 days from today's (ie 29-Nov-2021) date. Note: the ScholarOne system will 'lock' if submission of the revision is attempted 21 or more days after the deadline. If you do not think you will be able to meet this deadline please contact the editorial office immediately.

on behalf of Prof Marta Kwiatkowska (Subject Editor)
openscience@royalsociety.org

Associate Editor Comments to Author:

Thank you for your efforts to revise the paper and also for your patience while we sought additional reviewers for your paper - regrettably, two of the earlier reviewers were unable to re-review on this occasion, necessitating the search for replacements. This took longer than we would have preferred, and we offer apologies for any inconvenience caused.

Generally, the journal does not permit multiple rounds of revision, but given both the difficulties we've had in securing reviewers and the fact that one of the original reviewers has expressed satisfaction with your revisions, we'd like to offer you a final opportunity to revise the paper to tackle the concerns raised by the new reviewers. We are unlikely to be able to offer further rounds of revision (except for minor matters).

In addition to a number of other matters, two items that you should address appear pertinent: firstly, there is a question regarding the provision of the code (or code library) that supports your

work - please ensure this is fully accessible via a repository such as Zenodo before you resubmit. Secondly, there appears to be some dissonance between - on the one hand - your provision of supplementary material that does not directly identify the source materials and - on the other hand - yet includes sufficient metadata/identifiable material that the sources can be identified through a process of reverse-engineering. It would be worthwhile checking with the editorial office regarding this matter - given the journal's ethos of making data open, it may be possible to waive some of the confidentiality you've previously agreed to (this isn't guaranteed, but it would be sensible to ask).

Good luck and we'll look forward to receiving the revision in due course.

Reviewer comments to Author:

Reviewer: 1

Comments to the Author(s)

I am satisfied with the revised version of the paper. I consider the paper suitable for publication.

It is a pity that the identity of the four journals cannot be revealed. I wonder whether the Royal Society could give permission to the authors to reveal the identity of the journals. This information would substantially enrich the paper, while I believe it wouldn't harm the interests of the journals in any way.

Reviewer: 4

Comments to the Author(s)

The authors have access to an exciting data set, but sadly the quality of the paper does not live up to its promise. The main issue is the one raised previously by reviewer 2. It is not sufficiently clear why the authors choose the particular variables and engage in the particular analyses that they do. The introduction and discussion sections make bold claims about peer review and its purpose, but the connection with the novel content of the paper remains far too tacit. How should we interpret the number of statistical terms used in a given manuscript and how that changes in response to review? What is the connection with improving manuscripts (which the paper makes much of in the introduction and discussion)?

A minor comment: tables S3 and S4 seem to break the author's confidentiality agreement, as it is easy to tell from either of these tables that J1 is by far the largest journal and J8 the second largest. An informed reader could easily work out which Royal Society journals they are. This kind of disaggregated data by journal level should not be provided if the individual journals are meant to be anonymous. I'm not sure how to fix S3, but S4 could be easily fixed by focusing on the percentage of peer-reviewed manuscripts within each journal (i.e., making the columns add to 100% instead of the rows), which seems more relevant and interesting anyway.

Reviewer: 5

Comments to the Author(s)

Dear authors,

I read your manuscript with great interest, but I have some concerns. I know that this manuscript has been reviewed before and if you already addressed my comments to the satisfaction of other reviewers please let me know (plus tell me how you addressed them, please). As I understand it, your aim was to investigate the extent to which peer review affected the manuscripts' methodological rigor. As a proxy for methodological rigor, you used the number of statistical terms in the manuscripts. I have several concerns about the adequacy of this proxy and a number of things I could not glean from the text and would prefer to see clarified. I list these concerns and questions below.

My apologies for not being more positive and I hope my comments help in improving the manuscript.

Sincerely,
Noah van Dongen

Concerns about how you operationalize methodological rigor

Could you please explain why the number of statistical terms in a manuscript is a good indicator of methodological rigor. There might be very good reasons for using this proxy, but I did not see them expressed in the manuscript. I think the readers of your work need to see some kind of clarification on the relationship between methodological rigor and the number of statistical terms in a manuscript.

In general, research has been completed when the manuscript has been submitted for peer review. If this is so, could you please explain how peer review can affect the methodological rigor of the research? As I see it at the moment, and as long as no additional experiments are conducted, peer review can only affect the report quality of the research and the adequacy of the data processing and statistical analyses.

On page 3, you write that you generated six categories “four of them referring to various statistical approaches.” However, if I am not mistaken, in the rest of the manuscript no reference is made to these categories. It seems like only the total number of statistical terms are used and not the number of terms per category. If this is correct, I would like to see explained why this is. Also, are all the terms from the six categories used, or just the terms from the four categories referring to the statistical approaches?

Why are the absolute and not the relative numbers of statistical terms used? As I read the manuscript, you use counts of the identified statistical terms and not the number of terms with respect to, for instance, the length of the manuscript. Could you please provide some argumentation for this choice? I do not think you should categorize your manuscripts in ‘low’, ‘moderate’, and ‘high’ statistical content, if manuscripts can be of different lengths or can vary in the number of studies they report. Also, on occasion, you call manuscripts with low statistical content ‘manuscripts with weak statistical content.’ At the moment, I am missing the justification for using the term ‘weak.’

Some missing information I would like to see cleared up

How were the categories of ‘low’, ‘moderate’, and ‘high’ statistical content identified/defined? On page 6 you state that a moderate level of initial statistical content is “around 15 words”, but I did not read anything about how this was defined and why this is a moderate number of statistical terms (or in relation to what is this a moderate number).

When you mention that a certain percentage of manuscripts did not vary in statistical content between reviews, do you mean that the statistical terms are exactly the same, the number of terms is exactly the same, or the number of terms is approximately the same?

On page 7, you write “While these differences were on average minimal, we found cases where the number of statistical terms in the text greatly increased throughout a manuscript’s revision process.” What are you trying to convey in this sentence? Are you saying that there were outliers? I think this sentence needs specific numbers and/or further elaboration.

On page 7, you write about Figure 4 “the review score was increasingly instrumental for a manuscript’s final acceptance, yet the effect was not particularly prominent (top right)” As I read Figure 4, 'not particularly prominent' is an understatement. If I read the graph correctly, the probability of acceptance changed on average only a few percent from a reviewer score of zero to max. Is this correct?

The last sentence of the first paragraph of the discussion starts with “on the other hand”. I am sorry, but I do not understand to what this sentence is a contrast.

On page 7, you write “We found that these developments were instrumental to increase the probability of a manuscript’s acceptance.” I think most people would interpret the word ‘instrumental’ as the effect being both causal and strong/significant/extensive. Do you think the use of this term is warranted? Also, as I read it, (the analysis of) this causal connection is not specified in the manuscript. Could this be specified?

On page 7, you write “The availability of reviewer guidelines on statistics on review forms increased the similarity of statistical content among submitted manuscripts but did not have any qualitative effect on manuscript development during peer review.” Do you mean that different manuscripts were similar in their statistical content, or that each of these manuscripts did not change during the review process? Also, I don't see this result in the results section or in Figure 1. Or am I missing something?

Minor comments

In the last equation of page 4: should it not be $P(x_2 | D)$ (instead of X_1)?

On page 7, you write “four journals from The Royal Society over a similar time frame (2006-2017).” Don’t you mean ‘the same time frame’?

You end the manuscript with the limitations. Would it not be better to end with some general conclusion, suggestions for the future, or some kind of take-home-message?

===PREPARING YOUR MANUSCRIPT===

Your revised paper should include the changes requested by the referees and Editors of your manuscript. You should provide two versions of this manuscript and both versions must be provided in an editable format:
 one version identifying all the changes that have been made (for instance, in coloured highlight, in bold text, or tracked changes);
 a 'clean' version of the new manuscript that incorporates the changes made, but does not highlight them. This version will be used for typesetting if your manuscript is accepted.

If you have been asked to revise the written English in your submission as a condition of publication, you must do so, and you are expected to provide evidence that you have received language editing support. The journal would prefer that you use a professional language editing service and provide a certificate of editing, but a signed letter from a colleague who is a fluent speaker of English is acceptable. Note the journal has arranged a number of discounts for authors using professional language editing services (<https://royalsociety.org/journals/authors/benefits/language-editing/>).

===PREPARING YOUR REVISION IN SCHOLARONE===

Author's Response to Decision Letter for (RSOS-210681.R1)

See Appendix B.

RSOS-210681.R2

Review form: Reviewer 5 (Noah van Dongen)

Is the manuscript scientifically sound in its present form?

Yes

Are the interpretations and conclusions justified by the results?

No

Is the language acceptable?

Yes

Do you have any ethical concerns with this paper?

No

Have you any concerns about statistical analyses in this paper?

No

Recommendation?

Major revision is needed (please make suggestions in comments)

Comments to the Author(s)

The manuscript has greatly improved, but I still have a few comments. It could be that I comment on parts of the manuscript that were already present during the previous review round. My apologies for having missed them in my previous reading.

The main concern I still have is that a clear explanation is still missing of why the number of (different) statistical terms in a manuscript is a good proxy for methodological rigor. It is my

opinion that these sentences from the introduction are insufficient: “While claiming that any change of the statistical content of manuscripts during peer review would always lead to manuscript improvements in terms of methodological rigour could be questionable, mapping the outcome of a joint attention effort by reviewers and authors on the text of manuscripts would reveal a collective learning effort on the methodological content of manuscripts, which is one of the most important functions of peer review.” I think this is insufficient because the link between the (change in the) number of (different) statistical terms and the (improvement of) methodological rigor of the manuscript is not described and arguments are lacking why (a change in) the number of (different) statistical terms would reliably indicate the (improvement of) methodological rigor of the manuscript.

In section 2, I found the sentence “We assumed that the presence of statistical terms into the text would reflect statistical standards, concepts and knowledge incorporated in manuscripts, while making possible their full comparison (with other manuscripts and across journals)”. There should be a bit more argumentation about why you are allowed to assume this. (There are also two spelling mistakes in the sentence).

On page three, Section ‘(a) Data’, first paragraph, you state that you excluded manuscripts with missing files. However, you also state that you included “2,608 manuscripts without any available review (i.e., missing or not recorded in the journal submission system)”. How is this possible?

In Table 1, there are “Manuscripts with no review” (below “Peer reviewed manuscripts” and “Desk-rejection or acceptance”). Does this mean that they were actually reviewed, but no report was available? Or is there something else that can happen to a manuscript besides peer review and desk-decisions?

It is still not clear to me why you first go through creating five different categories of statistical terms and then lump them all together again for your analysis. Why put all this effort in and in addition make sure they don't overlap, if you are not using these categories? You should explain why create these categories in the first place and also why you then don't need to use these categories (but still needed to create them).

If it is indeed the case that you counted the number of different statistical terms in the manuscripts and reports (not the total number of statistical terms), then you should be consistent and state this everywhere you mention the number of statistical terms.

The word ‘linear’ can be removed from “Poisson linear regression” and “logistic linear regression”.

On page 7, you state “Generally, manuscripts with smaller changes in their statistical content were typically associated.” Could you be more specific?

On page 7, you write “Results showed that manuscripts eventually accepted for publication but receiving lower review scores were also those which increased their statistical content the most during peer review (see Figure 10 in the supplementary information).” This could be, but looking at the interquartile range in Figure 10, there is not much difference between the score of the paper that reduced their statistical content and the papers that increased their statistical content. Or am I misreading this? Could you please clarify or explain this?

On page 7, you write “Results of our models showed that the statistical content of a manuscript's final version was directly related to the level of statistical content of its initial version submitted for publication”. Does the word ‘directly’ mean that you tested and rejected the possibility of an indirect relation?

On page 7, you write "Figure 3 shows that the effect of the number of statistical terms in the reports and the number of rounds of peer review on the final statistical content of manuscripts is increasing and linear, although the greater changes during the process were mostly marginal, e.g., adding one more statistical term to the text." Do you mean adding on statistical term per review round? Or do you mean adding one statistical term between the first and last version of the manuscript?

On page 7, you write "For instance, the mean difference between versions was 1.9 of terms (median: 1) in case of manuscript acceptance, 2.1 (median: 2) in case of rejection. In case of manuscripts decreasing their statistical content under peer review, the mean difference was 1.8 fewer terms (median: 1), when the manuscript was then accepted for publication, 1.9 fewer terms (median: 1) whenever rejected. Although differences were small on average, we found outlier cases where the number of statistical terms in the text greatly increased throughout a manuscript's revision process." This seems very weird. It looks like if you take the manuscripts together, then on average there is no change in the number of different statistical terms in the manuscripts. Also, on average the accepted and rejected manuscripts change by the same amount (2). Thus, as I currently read it, these sentences don't convey much meaning (besides paper that increased, increased by some amount, and papers that decreased, decreased by about the same amount).

On page 8, you write "Our results indicated that manuscripts with both initial low or high levels of statistical content improved during the process, whereas desk-rejected manuscripts had comparatively fewer statistical terms in their text. We found that these developments had an influence on the probability of a manuscript's acceptance." However, on Figure 5 it appears that about 70% of the accepted papers either did not change in statistical content or changed by one term. Could you please clarify this? Should I be looking elsewhere for the results on which this conclusion is based?

Decision letter (RSOS-210681.R2)

Dear Professor Squazzoni,

The Editors assigned to your paper RSOS-210681.R2 "Does peer review improve the statistical content of manuscripts? A study on 27,467 manuscripts" have now received comments from reviewers and would like you to revise the paper in accordance with the reviewer comments and any comments from the Editors. We invite you to respond to the comments supplied below and revise your manuscript. Below the referees' and Editors' comments (where applicable) we provide additional requirements. Final acceptance of your manuscript is dependent on these requirements being met. We provide guidance below to help you prepare your revision.

Please submit your revised manuscript and required files (see below) no later than 21 days from today's (ie 23-May-2022) date. Note: the ScholarOne system will 'lock' if submission of the revision is attempted 21 or more days after the deadline. If you do not think you will be able to meet this deadline please contact the editorial office immediately.

on behalf of the Associate Editor and Professor Marta Kwiatkowska (Subject Editor)
openscience@royalsociety.org

Associate Editor Comments to Author:

The reviewer recognises the hard work put into this revision and, as their comments are between a minor and major revision, we've opted to give you the flexibility to complete the revisions in 3 weeks, rather than than 1, and to invite the reviewer to make a final assessment of the changes you make. We look forward to receiving your revised manuscript soon.

Reviewer comments to Author:

Reviewer: 5

Comments to the Author(s)

The manuscript has greatly improved, but I still have a few comments. It could be that I comment on parts of the manuscript that were already present during the previous review round. My apologies for having missed them in my previous reading.

The main concern I still have is that a clear explanation is still missing of why the number of (different) statistical terms in a manuscript is a good proxy for methodological rigor. It is my opinion that these sentences from the introduction are insufficient: "While claiming that any change of the statistical content of manuscripts during peer review would always lead to manuscript improvements in terms of methodological rigour could be questionable, mapping the outcome of a joint attention effort by reviewers and authors on the text of manuscripts would reveal a collective learning effort on the methodological content of manuscripts, which is one of the most important functions of peer review." I think this is insufficient because the link between the (change in the) number of (different) statistical terms and the (improvement of) methodological rigor of the manuscript is not described and arguments are lacking why (a change in) the number of (different) statistical terms would reliably indicate the (improvement of) methodological rigor of the manuscript.

In section 2, I found the sentence "We assumed that the presence of statistical terms into the text would reflect statistical standards, concepts and knowledge incorporated in manuscripts, while making possible their full comparison (with other manuscripts and across journals)". There

should be a bit more argumentation about why you are allowed to assume this. (There are also two spelling mistakes in the sentence).

On page three, Section '(a) Data', first paragraph, you state that you excluded manuscripts with missing files. However, you also state that you included "2,608 manuscripts without any available review (i.e., missing or not recorded in the journal submission system)". How is this possible?

In Table 1, there are "Manuscripts with no review" (below "Peer reviewed manuscripts" and "Desk-rejection or acceptance"). Does this mean that they were actually reviewed, but no report was available? Or is there something else that can happen to a manuscript besides peer review and desk-decisions?

It is still not clear to me why you first go through creating five different categories of statistical terms and then lump them all together again for your analysis. Why put all this effort in and in addition make sure they don't overlap, if you are not using these categories? You should explain why create these categories in the first place and also why you then don't need to use these categories (but still needed to create them).

If it is indeed the case that you counted the number of different statistical terms in the manuscripts and reports (not the total number of statistical terms), then you should be consistent and state this everywhere you mention the number of statistical terms.

The word 'linear' can be removed from "Poisson linear regression" and "logistic linear regression".

On page 7, you state "Generally, manuscripts with smaller changes in their statistical content were typically associated." Could you be more specific?

On page 7, you write "Results showed that manuscripts eventually accepted for publication but receiving lower review scores were also those which increased their statistical content the most during peer review (see Figure 10 in the supplementary information)." This could be, but looking at the interquartile range in Figure 10, there is not much difference between the score of the paper that reduced their statistical content and the papers that increased their statistical content. Or am I misreading this? Could you please clarify or explain this?

On page 7, you write "Results of our models showed that the statistical content of a manuscript's final version was directly related to the level of statistical content of its initial version submitted for publication". Does the word 'directly' mean that you tested and rejected the possibility of an indirect relation?

On page 7, you write "Figure 3 shows that the effect of the number of statistical terms in the reports and the number of rounds of peer review on the final statistical content of manuscripts is increasing and linear, although the greater changes during the process were mostly marginal, e.g., adding one more statistical term to the text." Do you mean adding on statistical term per review round? Or do you mean adding one statistical term between the first and last version of the manuscript?

On page 7, you write "For instance, the mean difference between versions was 1.9 of terms (median: 1) in case of manuscript acceptance, 2.1 (median: 2) in case of rejection. In case of manuscripts decreasing their statistical content under peer review, the mean difference was 1.8 fewer terms (median: 1), when the manuscript was then accepted for publication, 1.9 fewer terms (median: 1) whenever rejected. Although differences were small on average, we found outlier

cases where the number of statistical terms in the text greatly increased throughout a manuscript's revision process." This seems very weird. It looks like if you take the manuscripts together, then on average there is no change in the number of different statistical terms in the manuscripts. Also, on average the accepted and rejected manuscripts change by the same amount (2). Thus, as I currently read it, these sentences don't convey much meaning (besides paper that increased, increased by some amount, and papers that decreased, decreased by about the same amount).

On page 8, you write "Our results indicated that manuscripts with both initial low or high levels of statistical content improved during the process, whereas desk-rejected manuscripts had comparatively fewer statistical terms in their text. We found that these developments had an influence on the probability of a manuscript's acceptance." However, on Figure 5 it appears that about 70% of the accepted papers either did not change in statistical content or changed by one term. Could you please clarify this? Should I be looking elsewhere for the results on which this conclusion is based?

===PREPARING YOUR MANUSCRIPT===

If you have been asked to revise the written English in your submission as a condition of publication, you must do so, and you are expected to provide evidence that you have received language editing support. The journal would prefer that you use a professional language editing service and provide a certificate of editing, but a signed letter from a colleague who is a fluent speaker of English is acceptable. Note the journal has arranged a number of discounts for authors using professional language editing services (<https://royalsociety.org/journals/authors/benefits/language-editing/>).

===PREPARING YOUR REVISION IN SCHOLARONE===

Author's Response to Decision Letter for (RSOS-210681.R2)

See Appendix C.

RSOS-210681.R3

Review form: Reviewer 3 (Peiling Wang)

Is the manuscript scientifically sound in its present form?

Yes

Are the interpretations and conclusions justified by the results?

Yes

Is the language acceptable?

Yes

Do you have any ethical concerns with this paper?

No

Have you any concerns about statistical analyses in this paper?

Yes

Recommendation?

Accept with minor revision (please list in comments)

Comments to the Author(s)

This is a very interesting research with rarely accessible data. This paper focused on changes of statistical terms over the process of peer review.. With minor revisions, this paper should be published.

Here are a few points:

Please rephrase the last paragraph of Introduction to focus on what your research is about. This paragraph begins with "Claiming that any could be questionable" and reads confusing. It will be helpful to state clearly your research questions.

[I do not have access to the revision 2 and associated responses. In my record, I did suggest to clearly state research questions and hypotheses in my first review report.]

After accessing to the deposited data, I believe that the results should be definitely boiled down. The total number of records = 11,050 for the four journals measured by six variables:

Independent Variable: Number of reviewers

Independent Variable: Number of rounds of reviews

Independent Variable: Number of statistical Terms (initial submission)

Independent Variable: Max Change of statistical Terms

Independent Variable: Number of statistical Terms (final)

Dependent Variable: Manuscript decision (Accept vs. Reject)

Grouping variable: Journal

Page 4 is very hard to read with all the tracking and colored texts.

Page 5 presented model but I could not relate your variables with the model to understand how the results are derived. Please make "observing D" (page 10, line 50) concrete as which above variable is D; also, please be concrete on what variables in "After selecting these variables (page 6, line 19). I feel that these two pages read like teaching the formulas rather than how they are used in this study. It is import to show the use of the formulas with the variables and collected data.

For results why report manuscripts that were not peer reviewed? This is not relevant to your results also not in your analytical dataset. As I quote here "Figure 1 shows that initial submissions had a relatively homogeneous statistical content, except for manuscripts directly accepted by editors without any peer review (see the red solid line, which corresponded to 42 manuscripts)"? Should be deleted.

In Results, you stated "We then considered all 11,243 manuscripts" (page 11, line 56). and later on you explained why it was only 11,050. Let's make two points: first, the change from 11,243  11,050 be mentioned only once in Data section not in Results; second, it distracts or confuses readers at the time we should focus on the valid dataset.

The entire manuscript needs to boil down to minimize repetitive contents.

Hope your next revision is for readers rather than for reporting.

Review form: Reviewer 6

Is the manuscript scientifically sound in its present form?

Yes

Are the interpretations and conclusions justified by the results?

No

Is the language acceptable?

Yes

Do you have any ethical concerns with this paper?

No

Have you any concerns about statistical analyses in this paper?

No

Recommendation?

Major revision is needed (please make suggestions in comments)

Comments to the Author(s)

See Attached (Appendix D).

Decision letter (RSOS-210681.R3)

Dear Professor Squazzoni,

On behalf of the Editors, we are pleased to inform you that your Manuscript RSOS-210681.R3 "Does peer review improve the statistical content of manuscripts? A study on 27,467 manuscripts" has been accepted for publication in Royal Society Open Science subject to minor revision in accordance with the referees' reports. Please find the referees' comments along with any feedback from the Editors below my signature.

Please submit your revised manuscript and required files (see below) no later than 7 days from today's (ie 25-Jul-2022) date. Note: the ScholarOne system will 'lock' if submission of the revision is attempted 7 or more days after the deadline. If you do not think you will be able to meet this deadline please contact the editorial office immediately.

on behalf of Professor Marta Kwiatkowska (Subject Editor)
openscience@royalsociety.org

Associate Editor Comments to Author:

Thank you for your continued patience with the review of your paper. Unfortunately, in each round of review, we've been forced to seek at least one new reviewer to replace an earlier referee who has opted not to re-review. We're sorry for the delays this has imposed.

However, we would like you to make what we hope to be a final round of revisions, please, based on the remaining comments of these reviewers.

The Editors will then make decision whether the new reviewer needs to see the next version again or whether they can accept/reject finally on the basis of the rebuttals and revised manuscript you provide. Please do ensure that you do all you can to address their concerns, however.

Reviewer comments to Author:

Reviewer: 3

Comments to the Author(s)

Attached file: "royal society open.pdf"

This is a very interesting research with rarely accessible data. This paper focused on changes of statistical terms over the process of peer review.. With minor revisions, this paper should be published.

Here are a few points:

Please rephrase the last paragraph of Introduction to focus on what your research is about. This paragraph begins with "Claiming that any could be questionable" and reads confusing. It will be helpful to state clearly your research questions.

[I do not have access to the revision 2 and associated responses. In my record, I did suggest to clearly state research questions and hypotheses in my first review report.]

After accessing to the deposited data, I believe that the results should be definitely boiled down. The total number of records = 11,050 for the four journals measured by six variables:

Independent Variable: Number of reviewers

Independent Variable: Number of rounds of reviews

Independent Variable: Number of statistical Terms (initial submission)

Independent Variable: Max Change of statistical Terms

Independent Variable: Number of statistical Terms (final)

Dependent Variable: Manuscript decision (Accept vs. Reject)

Grouping variable: Journal

Page 4 is very hard to read with all the tracking and colored texts.

Page 5 presented model but I could not relate your variables with the model to understand how the results are derived. Please make "observing D" (page 10, line 50) concrete as which above variable is D; also, please be concrete on what variables in "After selecting these variables (page 6, line 19). I feel that these two pages read like teaching the formulas rather than how they are used in this study. It is import to show the use of the formulas with the variables and collected data.

For results why report manuscripts that were not peer reviewed? This is not relevant to your results also not in your analytical dataset. As I quote here "Figure 1 shows that initial submissions had a relatively homogeneous statistical content, except for manuscripts directly accepted by editors without any peer review (see the red solid line, which corresponded to 42 manuscripts)"? Should be deleted.

In Results, you stated "We then considered all 11,243 manuscripts" (page 11, line 56). and later on you explained why it was only 11,050. Let's make two points: first, the change from 11,243  11,050 be mentioned only once in Data section not in Results; second, it distracts or confuses readers at the time we should focus on the valid dataset.

The entire manuscript needs to boil down to minimize repetitive contents.

Hope your next revision is for readers rather than for reporting.

Reviewer: 6

Comments to the Author(s)

Attached file: "RSOS210681.JPG"

See Attached

===PREPARING YOUR MANUSCRIPT===

one version should clearly identify all the changes that have been made (for instance, in coloured highlight, in bold text, or tracked changes);

===PREPARING YOUR REVISION IN SCHOLARONE===

-- If you are requesting an article processing charge waiver, you must select the relevant waiver option (if requesting a discretionary waiver, the form should have been uploaded, see 'File upload' above).

-- If you have uploaded any electronic supplementary (ESM) files, please ensure you follow the guidance at <https://royalsociety.org/journals/authors/author-guidelines/#supplementary-material> to include a suitable title and informative caption. An example of appropriate titling and captioning may be found at https://figshare.com/articles/Table_S2_from_Is_there_a_trade-off_between_peak_performance_and_performance_breadth_across_temperatures_for_aerobic_scope_in_teleost_fishes_/3843624.

Author's Response to Decision Letter for (RSOS-210681.R3)

See Appendix E.

RSOS-210681.R4

Review form: Reviewer 6

Is the manuscript scientifically sound in its present form?

Yes

Are the interpretations and conclusions justified by the results?

Yes

Is the language acceptable?

Yes

Do you have any ethical concerns with this paper?

No

Have you any concerns about statistical analyses in this paper?

No

Recommendation?

Accept with minor revision (please list in comments)

Comments to the Author(s)

My concerns have been mostly addressed.

A few small suggestions for the title and abstract:

TITLE

1. I think using "manuscript" twice makes the title awkward. I would suggest something like "Does peer review improve the statistical content of manuscripts? A study of 27,467 submissions to 4 journals"
2. I don't think the data quite justify using "improve" and I would prefer "increase." But I'll let the editor decide if that's required.

ABSTRACT

1. "Peer review is instrumental for public trust.." The manuscript isn't about public trust, and peer review is instrumental for lots of other things, like its developmental function which is the one actually studied. I'd consider dropping this sentence.
2. Don't need the word "unluckily" and comes off as a bit subjective, maybe replace with "however" or "yet"
3. I'd consider using the word "amount" (of statistical content) somewhere here because the concept of "statistical content" is very abstract, and I think readers will think first about types of content rather than amount.
4. It's hard for a reader to know whether "low" and "high" cover the entire range of initial amounts, or if there's also a "medium" amount that the authors chose not to comment on, which makes it seem like they're downplaying null findings or something.

Decision letter (RSOS-210681.R4)

Dear Professor Squazzoni

On behalf of the Editors, we are pleased to inform you that your Manuscript RSOS-210681.R4 "Does peer review improve the statistical content of manuscripts? A study on 27,467 manuscripts" has been accepted for publication in Royal Society Open Science subject to minor revision in accordance with the referees' reports. Please find the referees' comments along with any feedback from the Editors below my signature.

Please submit your revised manuscript and required files (see below) no later than 7 days from today's (ie 22-Aug-2022) date. Note: the ScholarOne system will 'lock' if submission of the revision is attempted 7 or more days after the deadline. If you do not think you will be able to meet this deadline please contact the editorial office immediately.

on behalf of Prof Marta Kwiatkowska (Subject Editor)
openscience@royalsociety.org

Associate Editor Comments to Author:

The remaining comments from the reviewer may be helpful to consider, but the Editors do not consider that any further review will be necessary after you submit a final version - thank you for your patience with the unusually lengthy review process on this occasion. Please note that it is the view of the Editors that 'improve' (rather than 'increase') is acceptable in the paper title.

Reviewer comments to Author:

Reviewer: 6

Comments to the Author(s)

My concerns have been mostly addressed.

A few small suggestions for the title and abstract:

TITLE

1. I think using "manuscript" twice makes the title awkward. I would suggest something like "Does peer review improve the statistical content of manuscripts? A study of 27,467 submissions to 4 journals"
2. I don't think the data quite justify using "improve" and I would prefer "increase." But I'll let the editor decide if that's required.

ABSTRACT

1. "Peer review is instrumental for public trust.." The manuscript isn't about public trust, and peer review is instrumental for lots of other things, like its developmental function which is the one actually studied. I'd consider dropping this sentence.
2. Don't need the word "unluckily" and comes off as a bit subjective, maybe replace with "however" or "yet"
3. I'd consider using the word "amount" (of statistical content) somewhere here because the concept of "statistical content" is very abstract, and I think readers will think first about types of content rather than amount.
4. It's hard for a reader to know whether "low" and "high" cover the entire range of initial amounts, or if there's also a "medium" amount that the authors chose not to comment on, which makes it seem like they're downplaying null findings or something.

===PREPARING YOUR MANUSCRIPT===

one version should clearly identify all the changes that have been made (for instance, in coloured highlight, in bold text, or tracked changes);
 a 'clean' version of the new manuscript that incorporates the changes made, but does not highlight them. This version will be used for typesetting.

If you have been asked to revise the written English in your submission as a condition of publication, you must do so, and you are expected to provide evidence that you have received language editing support. The journal would prefer that you use a professional language editing service and provide a certificate of editing, but a signed letter from a colleague who is a proficient user of English is acceptable. Note the journal has arranged a number of discounts for authors

using professional language editing services
(<https://royalsociety.org/journals/authors/benefits/language-editing/>).

===PREPARING YOUR REVISION IN SCHOLARONE===

-- If you are requesting an article processing charge waiver, you must select the relevant waiver option (if requesting a discretionary waiver, the form should have been uploaded, see 'File upload' above).

-- If you have uploaded any electronic supplementary (ESM) files, please ensure you follow the guidance at <https://royalsociety.org/journals/authors/author-guidelines/#supplementary-material> to include a suitable title and informative caption. An example of appropriate titling and

captioning may be found at https://figshare.com/articles/Table_S2_from_Is_there_a_trade-off_between_peak_performance_and_performance_breadth_across_temperatures_for_aerobic_sc_ope_in_teleost_fishes_/3843624.

Author's Response to Decision Letter for (RSOS-210681.R4)

See Appendix F.

Decision letter (RSOS-210681.R5)

Dear Professor Squazzoni:

I am pleased to inform you that your manuscript entitled "Does peer review improve the statistical content of manuscripts? A study on 27,467 submissions to four journals" is now accepted for publication in Royal Society Open Science.

Please remember to make any data sets or code libraries 'live' prior to publication, and update any links as needed when you receive a proof to check - for instance, from a private 'for review' URL to a publicly accessible 'for publication' URL. It is also good practice to add data sets, code and other digital materials to your reference list.

Royal Society Open Science is a fully open access journal. A payment may be due before your article is published. Our partner Copyright Clearance Center's RightsLink for Scientific Communications will contact the corresponding author about your open access options from the email domain @copyright.com (if you have any queries regarding fees, please see <https://royalsocietypublishing.org/rsos/charges> or contact authorfees@royalsociety.org).

Please see the Royal Society Publishing guidance on how you may share your accepted author manuscript at <https://royalsociety.org/journals/ethics-policies/media-embargo/>. After publication, some additional ways to effectively promote your article can also be found here

<https://royalsociety.org/blog/2020/07/promoting-your-latest-paper-and-tracking-your-results/>.

on behalf of Professor Marta Kwiatkowska (Subject Editor).

Follow Royal Society Publishing on Twitter: @RSocPublishing
Follow Royal Society Publishing on Facebook:
<https://www.facebook.com/RoyalSocietyPublishing/>
Read Royal Society Publishing's blog:
<https://royalsociety.org/blog/blogsearchpage/?category=Publishing>

Appendix A

Reviewer: 1

I find the following statement in the introduction quite questionable: “the supreme function of reviewers is to help journals achieve the highest methodological rigour and statistical standards of scientific publications”. Peer review and peer reviewers have many different functions. Improving methodological and statistical quality is one of them, but there are many others as well (e.g., improving the conceptual design of the research under review, improving the interpretation of empirical findings, improving the embedding in the literature, improving the presentation of the research under review, assessing the relevance of the article under review for a particular research community, advising a journal editor on the suitability of the article under review for publication in a journal, etc.).

Thank you for this comment. You are right. Reviewers have multiple functions. We have specified this better in the text.

The authors do not reveal the identity of the four journals included in the analysis (although they do mention that the journals are published by the Royal Society). Different research fields use different scientific methods. Statistical methods are just one type of scientific method. There are many others as well (e.g., machine learning, computer simulation, agent-based modeling, mathematical modeling, qualitative research methods, etc.). Since the identity of the four journals is not revealed, it is unclear to what extent these journals publish research from fields that do indeed rely strongly on statistical methods. It is therefore hard to assess whether the focus of the authors on statistical methods (as opposed to other types of scientific methods) is a sensible choice. Knowing the identity of the four journals would be highly informative for readers of the paper. I therefore hope the authors are able to reveal the identity of the journals. If this is not possible, the authors should at least provide a brief explanation of the research fields on which each of the four journals is focused and of the different scientific methods used in these research fields.

Making journal identity confidential was part of the agreement we all signed with The Royal Society to access internal journal data. However, given that we restricted our observational sample to only four journals to ensure comparability, we can share some detail on all RS journals without violating such an agreement. We therefore have improved the part in which we describe the journals. Thank you for this comment

For many of the results discussed in the results section, the actual results are not

shown in this section. Instead, they are shown in tables and figures that are made available as supplementary information. This makes the results section hard to read. It would be much easier for the reader if all results, or at least most of them, are shown in the paper itself rather than in the supplementary information.

We have followed the journal instructions and re-structured the paper, reducing the size of the supporting information and expanding on the Methods.

The figures are difficult to read. The authors should use a much larger font size.

We have improved the quality of all figures.

“We concentrated here on ... in published manuscripts”: Generalizing from ‘statistical content’ to ‘quality and rigour of the analysis’ is quite a big step. The authors need to reflect more carefully on this. To what extent does ‘statistical content’ indeed offer a reasonable proxy of ‘quality and rigour of the analysis’? I believe that statistical content is an important element of quality and rigour (at least in some research fields), but various other elements are important as well (e.g., design of the analysis, data collection approach, interpretation of the results, etc.), and these elements are not covered by the analysis of the authors.

You are right. We have added a point on this in the limitations section at the end of the manuscript.

“the most common terms in all statistical methodologies”: This is too strong and needs to be rephrased (e.g., “commonly used statistical terms”). Various statistical methodologies are not covered by the terms in Table 2, for instance commonly used multivariate analysis methods such as multidimensional scaling, principal component analysis, factor analysis, correspondence analysis, etc., statistical learning methods such as k-means clustering, hierarchical clustering, discriminant analysis, etc., and probably many other methodologies as well.

You are right. We have rephrased this.

“To understand which variables needed to be included”: Please list the variables that are taken into consideration.

These variables are listed in Tabela S1 and S2. We mentioned this in the text.

Reviewer: 2

Comments to the Author(s)

My own expertise lies in the humanistic and social sciences, not in computer science nor statistics. I can see that there has been substantial work done to calculate the number of statistical terms used in tens of thousands of papers and reviews, to calculate their variation, and the changes made during the review process. But what does it all mean? Is this statistical analysis answering an interesting or relevant question?

Thank you for this comment. The relevance of our work lies in the fact that we tried to measure the effect of reviewers in the change of manuscript. Obviously, our study is limited to a specific aspect, i.e., the statistical content. However, there is no previous research on this important function of peer review. We have specified this among the study limitations.

For this paper to be interesting to the wider community of those studying peer review, it would need more discussion/explanation of the measure used, what it might actually mean, and what this tells us about the role of statistics in scientific papers in the mind of authors and reviewers. At present, the 'discussion' (section 3) doesn't add very much to the description of results (section 2).

Among the questions that I had were:

- Why is 'number of statistical terms mentioned' assumed to be a meaningful measure of statistical content? [for instance, why is it more meaningful than 'proportion of the article devoted to graphs and/or paragraphs of text with statistical terms in them'?).

Thank you for this comment. The proportion of text changed is captured in our measurements as do all tables that were formatted as text. We excluded figures as they were objects and not text. Counting graphs is possible but distinguishing the ones reporting statistical analysis from those including visualisations, diagrams or any other element was impossible.

- What is the justification for assuming that 'more statistical terms mentioned' = 'better', and 'fewer terms' = 'weak' [as for J8]? This seems like a tacit assumption coming from authors based in a statistics-heavy discipline. For instance, there are clear disciplinary variations in the amount of statistical content that is required/expected,

and the Royal Society does publish journals with content from a wide variety of disciplines. So, what makes the statistical content of J8 'weak' rather than 'of a different discipline'? And would a very long paragraph full of statistical terms that was, nonetheless, complete rubbish still count as 'better' because it has more hits for the key terms?

Thank you for this comment. Of course, you are completely right. Paradoxically, a paper can increase its statistical content from round 1 to round 2 of peer review because reviewers contested exactly the quality of a given statistical test. However, note that we never state that increasing the content is good or bad, but it's a sign that reviewers and authors are concentrated on these aspects. In the revised version, we added details on journals so that the reader can see that journals were relatively homogeneous in terms of scientific standards. We also added a couple of sentences in the closing section to clarify the study limitations better.

- Did submitted papers with lots of statistical content generate reviewer reports with lots of commentary on statistics? And did submitted papers with little statistical content generate reviewer reports with lots of comments on statistics (implying: more statistics required) or not (implying: this is normal in this field)?

Thank you for this comment. The Pearson correlation between the initial number of statistical terms in a manuscript and the maximum number of review terms in the first round was 0.34, so weakly correlated. In any case, this correlation has poor meaning for the rationale of your study, which was to estimate the effect of peer review on changes of statistical content. In our models, we considered the effect of the initial level of statistical terms in the manuscript.

- How can we know whether reviewer reports with lots of statistical terms in them were critiquing or appreciating the statistics in the submitted paper?

Thank you for this comment. We clearly could not distinguish between positive or critical comments and conversations. We only measured the fact that reviewers were pointing their attention to statistics and related concepts in their comments. A nice idea would be to extrapolate a sample of reports and dig into their text to adjudicate about the positive/negative comments therein. However, this would then pose the problem as to how relating these dummy (negative vs. positive) to our outcome measurements, how to balance a list of positive/negative comments in the same reports (e.g., averaging negative and positive and assigning a given outcome). In short, a great and

stimulating question, which unfortunately cannot be convincingly measured and would require a qualitative study on a much less ample sample of reports.

- Did the three journals with guidance to reviewers asking them to look at statistics have that guidance in place throughout the time period studied? [2006 seems quite early to have that in place...] Did the introduction of such guidance lead to any change of behaviour among reviewers? Or among authors making revisions?

Journals having guidelines did have them during our observation. We disentangled the effect of guidelines from non-guidelines in our analysis and found weak effects – as reported in the text.

Reviewer: 3

Comments to the Author(s)

The first issue is the structure. I wondered why the authors deviated from the instruction to the author and the journal-provided template?

Currently, the Results and Discussion sections preceded the Methods. The Conclusions section is missing. This unusual order of contents made the manuscript hard to follow. Please use the template at

https://royalsocietypublishing.org/pb-assets/RSOS%20assets/RSOS_template-1580479300553.docx

We restructured the text. This also implied reducing the size of the supporting information and expand on the Methods section, as suggested by another reviewer.

I suggest providing a conceptual framework to relate specific research questions with the statistical analysis. Please state clearly the hypotheses to connect variables (such as dependent with independent) in order to justify the chosen statistics, if the intention was to test if there was a significant effect on ... (e.g., Page 5, Line 4-5; Page 9, Line 42)

Thank you for this comment. We believe that we have justified the choice of our statistical models better now.

In terms of the first step of data processing the six categories of LIWC, I wondered if the authors did manually or applied a computer program (Page 9, Line 15-). Had the processed data been illustrated, readers could perhaps make sense of the analysis or

results. I tried to open the dataset following the link but, the link is inactive in either the html or the pdf version:

*Sorry to hearing that you had problems accessing our dataset. We double checked the link and it worked. However, we have shifted it to a public one now with a DOI: <https://doi.org/10.7910/DVN/MOKJED>. The link has also been added to the manuscript. The dictionary can be downloaded in the LIWC format and anyone can use it either in the LIWC software or in any available Python/R libraries. As explained in the text, we used “*quanteda.dictionaries*”, an R library to extract the statistical content as the sum of the number of statistical terms in each category.*

B Data availability

The dataset used for this study is available to reviewers here.

I am researching peer review using OPR. Therefore, I have basic ideas of what review reports are. But the authors did not share or illustrate how they coded data using the six categories. Some specifics on methods were missing.

Thank you for this comment. We have improved the description of our methods in the manuscript. As mentioned above, we have all data in a public repository for replication.

Although valuable work, the authors did not write according to the journal's guidelines, nor following statistical convention: Research questions  hypothesis 

Thank you for this comment. We have now restructured the manuscript following the journal instructions more carefully. However, note that our study was exploratory. We did not follow a strict hypothesis-testing design.

Appendix B

Reviewer comments to Author:

Reviewer: 1

Comments to the Author(s)

I am satisfied with the revised version of the paper. I consider the paper suitable for publication. It is a pity that the identity of the four journals cannot be revealed. I wonder whether the Royal Society could give permission to the authors to reveal the identity of the journals. This information would substantially enrich the paper, while I believe it wouldn't harm the interests of the journals in any way.

Thank you for your comment. We have added some detail on the journals. However, maintaining journal anonymity is key not only to comply with previously established agreements and not only for this specific work, for which - we agree – there is no reputational risk involved for the publisher. Given the usual problems in convincing publishers to share journal data, most conservative publishers could anticipate possible reputational risk from non-anonymization (surely over-exaggerated) in current research to defend their decision to not to share their internal data with the academic community in the future.

Reviewer: 4

Comments to the Author(s)

The authors have access to an exciting data set, but sadly the quality of the paper does not live up to its promise. The main issue is the one raised previously by reviewer 2. It is not sufficiently clear why the authors choose the particular variables and engage in the particular analyses that they do. The introduction and discussion sections make bold claims about peer review and its purpose, but the connection with the novel content of the paper remains far too tacit. How should we interpret the number of statistical terms used in a given manuscript and how that changes in response to review? What is the connection with improving manuscripts (which the paper makes much of in the introduction and discussion)?

Thank you for this suggestion. We have added some sentences at the end of the intro paragraph to outline the rationale behind the work. We also cited a recently published manuscript that addresses the importance of linguistic analysis of peer review.

A minor comment: tables S3 and S4 seem to break the author's confidentiality agreement, as it is easy to tell from either of these tables that J1 is by far the largest journal and J8 the second largest. An informed reader could easily work out which Royal Society journals they are. This kind of disaggregated data by journal level should not be provided if the individual journals are meant to be anonymous. I'm not sure how to fix S3, but S4 could be easily fixed by focusing on the percentage of peer-reviewed manuscripts within each journal (i.e., making the columns add to 100% instead of the rows), which seems more relevant and interesting anyway.

Thank you for these suggestions. We have reconsidered these tables. Table 5 in the supplementary material could help more informed readers to identify from the distribution given, the specific journals behind the J1, 2, ... classification. However, we do not find this problematic as any analysis and conclusion regarding our findings make the link to any specific journal difficult to reconstruct.

Reviewer: 5

Comments to the Author(s)

Dear authors,

I read your manuscript with great interest, but I have some concerns. I know that this manuscript has been reviewed before and if you already addressed my comments to the satisfaction of other reviewers please let me know (plus tell me how you addressed them, please). As I understand it, your aim was to investigate the extent to which peer review affected the manuscripts' methodological rigor. As a proxy for methodological rigor, you used the number of statistical terms in the manuscripts. I have several concerns about the adequacy of this proxy and a number of things I could not glean from the text and would prefer to see clarified. I list these concerns and questions below.

My apologies for not being more positive and I hope my comments help in improving the manuscript.

Sincerely,
Noah van Dongen

Concerns about how you operationalize methodological rigor

Could you please explain why the number of statistical terms in a manuscript is a good indicator of methodological rigor. There might be very good reasons for using this proxy, but I did not see them expressed in the manuscript. I think the readers of your work need to see some kind of clarification on the relationship between methodological rigor and the number of statistical terms in a manuscript.

We have added a paragraph at the end of the intro section and one in Section 2 when we presented our dataset in which we clarified better our assumptions.

In general, research has been completed when the manuscript has been submitted for peer review. If this is so, could you please explain how peer review can affect the methodological rigor of the research? As I see it at the moment, and as long as no additional experiments are conducted, peer review can only affect the report quality of the research and the adequacy of the data processing and statistical analyses.

Of course, examining text change during peer review, as well as the link between text of reviewer reports and manuscript text change, is only a proxy of the potential improvement of the methodological rigour of manuscripts. However, measuring initial

content vs. changes is key to estimate the effect of peer review, which was the goal of the study. This said, it is hard to reconstruct quantitatively the effect of peer review besides exactly what you mentioned in the final sentence: “peer review can only affect the report quality of the research and the adequacy of the data processing and statistical analyses”. This is the point: authors can modify, run further test, reconsider references, improving the clarity of the text under recommendations by reviewers. It is obviously rare that they re-design the initial study but we believe this is not the function of peer review.

On page 3, you write that you generated six categories “four of them referring to various statistical approaches.” However, if I am not mistaken, in the rest of the manuscript no reference is made to these categories. It seems like only the total number of statistical terms are used and not the number of terms per category. If this is correct, I would like to see explained why this is. Also, are all the terms from the six categories used, or just the terms from the four categories referring to the statistical approaches?

Many thanks for pointing this out. We have structured the terms in our dictionary into five categories (see Table 2), which helped us to conceptualize the statistical approach they could refer to. We have also checked for between-terms orthogonality in order to ensure that each term represented different concepts with no overlapping. To analyze the presence of any of these concepts, we extracted the overall statistical content of manuscripts in terms of the number of different statistical terms belonging to any category. Note that we have aggregated all categories since our focus was the statistical content as a whole, regardless of the nature of manuscripts or concepts (either descriptive, inferential or both). We have revised the text to better explain this.

Why are the absolute and not the relative numbers of statistical terms used? As I read the manuscript, you use counts of the identified statistical terms and not the number of terms with respect to, for instance, the length of the manuscript. Could you please provide some argumentation for this choice?

Thanks for this comment. As mentioned in our previous response, we actually counted the number of statistical terms found in the text but we only kept the first occurrence as we were just checking for the presence of such statistical concepts. Our ‘whole’ was then the total number of terms in our dictionary, which was constant and did not vary depending on the manuscript length. Therefore, the analysis on this absolute number must be seen as equivalent to using the relative number.

I do not think you should categorize your manuscripts in ‘low’, ‘moderate’, and ‘high’ statistical content, if manuscripts can be of different lengths or can vary in the number of studies they report. Also, on occasion, you call manuscripts with low statistical content ‘manuscripts with weak statistical content.’ At the moment, I am missing the justification for using the term ‘weak.’

We agree on the inappropriate use of the term “weak”. We now used “low”, “moderate” and “high” consistently across the paper. By using these terms, we were

referring to our measure of the statistical content of manuscripts, which – as explained above - did not depend on manuscript length. In Figure 2, we showed how this could vary depending on the study reported and the statistical content of review reports.

Some missing information I would like to see cleared up

How were the categories of ‘low’, ‘moderate’, and ‘high’ statistical content identified/defined? On page 6 you state that a moderate level of initial statistical content is “around 15 words”, but I did not read anything about how this was defined and why this is a moderate number of statistical terms (or in relation to what is this a moderate number).

We used “low”, “moderate” and “high” to reflect the amount of statistical content with respect to the maximum number of statistical terms, which as around 30 terms as shown in Figure 1. We have added a comment on this in the text that clarified this point.

When you mention that a certain percentage of manuscripts did not vary in statistical content between reviews, do you mean that the statistical terms are exactly the same, the number of terms is exactly the same, or the number of terms is approximately the same?

Many thanks for this comment. By this, we meant that the number of different terms from the dictionary found in the manuscript was exactly the same. We have added a sentence in the second paragraph of section Results to clarify this.

On page 7, you write “While these differences were on average minimal, we found cases where the number of statistical terms in the text greatly increased throughout a manuscript’s revision process.” What are you trying to convey in this sentence? Are you saying that there were outliers? I think this sentence needs specific numbers and/or further elaboration.

Thanks. Yes, we were referring to certain outliers. We revised the sentence.

On page 7, you write about Figure 4 “the review score was increasingly instrumental for a manuscript’s final acceptance, yet the effect was not particularly prominent (top right)” As I read Figure 4, 'not particularly prominent' is an understatement. If I read the graph correctly, the probability of acceptance changed on average only a few percent from a reviewer score of zero to max. Is this correct?

Good point. The review score was influential for a manuscript's final acceptance as it was closely related with the number of rounds. For instance, manuscripts undergoing only one round of peer review had a median review score of 0.12 whereas those undergoing more than one round had a median review score of 0.67. The effect of the number of rounds is somehow confounded with that of the review score, as shown in

the marginal effect for papers having two review rounds (top right of Figure 4). However, we kept both variables to build a better model as indicated by their posterior inclusion probabilities (as suggested from Table 4). We have revised the text and the caption of Figure 4 to outline this better.

The last sentence of the first paragraph of the discussion starts with “on the other hand”. I am sorry, but I do not understand to what this sentence is a contrast.

Thank you for this remark. We revised the sentence.

On page 7, you write “We found that these developments were instrumental to increase the probability of a manuscript’s acceptance.” I think most people would interpret the word ‘instrumental’ as the effect being both causal and strong/significant/extensive. Do you think the use of this term is warranted? Also, as I read it, (the analysis of) this causal connection is not specified in the manuscript. Could this be specified?

Thank you for this remark. We revised the sentence.

On page 7, you write “The availability of reviewer guidelines on statistics on review forms increased the similarity of statistical content among submitted manuscripts but did not have any qualitative effect on manuscript development during peer review.” Do you mean that different manuscripts were similar in their statistical content, or that each of these manuscripts did not change during the review process? Also, I don't see this result in the results section or in Figure 1. Or am I missing something?

This is a point for you guys to answer

Minor comments

In the last equation of page 4: should it not be $P(x_2|D)$ (instead of X_1)?

Thanks. Typo fixed.

On page 7, you write “four journals from The Royal Society over a similar time frame (2006- 2017).” Don’t you mean ‘the same time frame’?

Corrected.

You end the manuscript with the limitations. Would it not be better to end with some general conclusion, suggestions for the future, or some kind of take-home-message?

We have revised the last paragraphs.

Appendix C

RSOS-210681.R2 "Does peer review improve the statistical content of manuscripts? A study on 27,467 manuscripts"

Reviewer: 5

Comments to the Author(s)

The manuscript has greatly improved, but I still have a few comments. It could be that I comment on parts of the manuscript that were already present during the previous review round. My apologies for having missed them in my previous reading.

The main concern I still have is that a clear explanation is still missing of why the number of (different) statistical terms in a manuscript is a good proxy for methodological rigor. It is my opinion that these sentences from the introduction are insufficient: "While claiming that any change of the statistical content of manuscripts during peer review would always lead to manuscript improvements in terms of methodological rigour could be questionable, mapping the outcome of a joint attention effort by reviewers and authors on the text of manuscripts would reveal a collective learning effort on the methodological content of manuscripts, which is one of the most important functions of peer review." I think this is insufficient because the link between the (change in the) number of (different) statistical terms and the (improvement of) methodological rigor of the manuscript is not described and arguments are lacking why (a change in) the number of (different) statistical terms would reliably indicate the (improvement of) methodological rigor of the manuscript.

Thank you for this remark. We agree on the fact that measuring the statistical improvement of a manuscript by counting statistical terms in reviews and pre-post manuscript version is not a perfect measurement. However, even if we would have a better measurement than terms, e.g., changes in table values or adding new tables (new results) across different versions of the manuscript, our capacity of assessing the methodological rigour would be questionable. Assessing methodological rigour would require a more qualitative study and in any case, it would be controversial. Indeed, measuring methodological rigour was not our goal. In any case, we have revised the sentence and clarifies our study's goal better. We also added a sentence on this in the limitations at the end of the paper.

In section 2, I found the sentence "We assumed that the presence of statistical terms into the text would reflect statistical standards, concepts and knowledge incorporated in manuscripts, while making possible their full comparison (with other manuscripts and across journals)". There should be a bit more argumentation about why you are

allowed to assume this. (There are also two spelling mistakes in the sentence).

Thank you. Sentence revised. We believe that the answer to the previous comment applies to this, too.

On page three, Section '(a) Data', first paragraph, you state that you excluded manuscripts with missing files. However, you also state that you included "2,608 manuscripts without any available review (i.e., missing or not recorded in the journal submission system)". How is this possible?

Manuscript files included all manuscript contents (e.g., version 1 = initial submission; version 2 = revised submission etc.) + associated review reports. In some cases, we did not have review reports but we still performed analysis on their changes

In Table 1, there are "Manuscripts with no review" (below "Peer reviewed manuscripts" and "Desk-rejection or acceptance"). Does this mean that they were actually reviewed, but no report was available? Or is there something else that can happen to a manuscript besides peer review and desk-decisions?

Correct. As mentioned in the text, in some cases (fortunately, not many), we could not find reports linked to these manuscripts.

It is still not clear to me why you first go through creating five different categories of statistical terms and then lump them all together again for your analysis. Why put all this effort in and in addition make sure they don't overlap, if you are not using these categories? You should explain why create these categories in the first place and also why you then don't need to use these categories (but still needed to create them).

Thank you. This was an explorative study. We reported all steps in the creation of the dataset.

If it is indeed the case that you counted the number of different statistical terms in the manuscripts and reports (not the total number of statistical terms), then you should be consistent and state this everywhere you mention the number of statistical terms.

Thank you. You right. We have revised the text, which is now consistent.

The word 'linear' can be removed from "Poisson linear regression" and "logistic linear

regression”.

Done. Thank you.

On page 7, you state “Generally, manuscripts with smaller changes in their statistical content were typically associated.” Could you be more specific?

Good point. We added detail on Figure 9 mentioned in the text.

On page 7, you write “Results showed that manuscripts eventually accepted for publication but receiving lower review scores were also those which increased their statistical content the most during peer review (see Figure 10 in the supplementary information).” This could be, but looking at the interquartile range in Figure 10, there is not much difference between the score of the paper that reduced their statistical content and the papers that increased their statistical content. Or am I misreading this? Could you please clarify or explain this?

When looking at inter-quartile ranges of figure 10, you can find in any case certain differences. However, note that these figures were mostly descriptive. To assess statistical significance and the size of effects, we have developed more rigorous models in the paper.

On page 7, you write “Results of our models showed that the statistical content of a manuscript’s final version was directly related to the level of statistical content of its initial version submitted for publication”. Does the word ‘directly’ mean that you tested and rejected the possibility of an indirect relation?

Good point. We have revised the text.

On page 7, you write “Figure 3 shows that the effect of the number of statistical terms in the reports and the number of rounds of peer review on the final statistical content of manuscripts is increasing and linear, although the greater changes during the process were mostly marginal, e.g., adding one more statistical term to the text.” Do you mean adding on statistical term per review round? Or do you mean adding one statistical term between the first and last version of the manuscript?

We meant statistical terms in the text, i.e., manuscripts.

On page 7, you write “For instance, the mean difference between versions was 1.9 of terms (median: 1) in case of manuscript acceptance, 2.1 (median: 2) in case of rejection. In case of manuscripts decreasing their statistical content under peer review, the mean difference was 1.8 fewer terms (median: 1), when the manuscript was then accepted for publication, 1.9 fewer terms (median: 1) whenever rejected. Although differences were small on average, we found outlier cases where the number of statistical terms in the text greatly increased throughout a manuscript’s revision process.” This seems very weird. It looks like if you take the manuscripts together, then on average there is no change in the number of different statistical terms in the manuscripts. Also, on average the accepted and rejected manuscripts change by the same amount (2). Thus, as I currently read it, these sentences don’t convey much meaning (besides paper that increased, increased by some amount, and papers that decreased, decreased by about the same amount).

Good point. We have removed this sentence.

On page 8, you write “Our results indicated that manuscripts with both initial low or high levels of statistical content improved during the process, whereas desk-rejected manuscripts had comparatively fewer statistical terms in their text. We found that these developments had an influence on the probability of a manuscript’s acceptance.” However, on Figure 5 it appears that about 70% of the accepted papers either did not change in statistical content or changed by one term. Could you please clarify this? Should I be looking elsewhere for the results on which this conclusion is based?

Figure 2 shows the dynamics we mentioned in the text, which has been confirmed by our Bayesian model. Figure 5 shows that changes are marginal. You are correct. We have revised the sentence and the text on results to tone down the style.

Appendix D

Review of “Does peer review improve the statistical content of manuscripts?”

This manuscript tries to assess whether peer review improves the “statistics” (leaving this purposefully vague for now) in research papers, using a sizable sample of manuscripts submitted to 4 Royal Society journals. In general, we have shockingly little evidence on what, if anything, peer review improves, so the question the authors ask is an important one. The biggest issue I see is with the key measure: statistical content. I’ll comment on that first and then some more medium/minor issues.

Statistical content

I think this is a nice phrase for what to call the measure and it feels appropriately close to the data. However, if no additional analyses/measures are provided, then the framing of this measure should be kept very neutral, *i.e.* the authors should speak of statistical content increasing or decreasing, but not “improving.” I think to frame the changes in content as “improvement” would require more direct measures. Perhaps something like <https://en.wikipedia.org/wiki/Statcheck> or similar tools, if there are any, can be used here?

So in my view, the authors face a choice between 1) finding more direct measures of “improvement” (maybe even on a small scale) and then showing that the amount of statistical content correlates with the quality of statistical content or 2) reframing the findings as about peer review increasing the amount of statistical content and speculating, with substantial hedging. At the very least “improve” should be removed from the title, and, if authors insist on having it in the abstract, it should be something like “peer review increases content...which may signal improvement.”

I agree with Reviewer 5 that the current justification for the measure as a proxy for statistical quality in the last paragraph before “2. Methods” is insufficient. In fact, I would probably remove that paragraph altogether. I would consider replacing it with reasons we might think that amount of statistical content equal quality of content. For example, I suspect there is literature showing that much statistical analysis is not reported in sufficient detail, so that’s at least one case where more content is widely believed to make manuscripts better. There are also lots of discussions that peer reviewers “force” authors to conduct (unnecessary?) additional experiments and robustness tests. So that’s another way for statistical content to increase in an (arguably) beneficial way.

Medium issues

- There is a substantial amount of causal language, particularly using the word “effects” to discuss associations, that should be phrased as associations. For example, on page 11, the authors say “The availability of guidelines ... did not have any qualitative effect on the variation...” but ought to say something like “The availability of guidelines ... was not associated with....”
- I think the authors should give more weight in the Discussion and maybe Introduction to the possibility that amount of statistical content is a bad measure of quality, and/or peer review does not improve quality all that much. There are a few pieces of data that point in that direction:
 - Figure 5: In a substantial number of cases, papers that were accepted decreased in statistical content. If more content is better, why would it decrease so often? Perhaps

amount of content is a proxy for number of analyses, and reducing these helps focus the paper on a key message.

- Figure 5: Among rejected papers, there was almost no change in statistical content. If peer review improves content, why would that only appear for accepted papers? If the answer is that the *published* version is necessary for these effects to be visible, then could it be due to copy-editing step rather than peer review?
- Overall, the changes in amount of terms seem quite small, so that is worth commenting upon. It seems there is an error in the first paragraph on page 13, “Figure 3 shows...”. I assume the authors want to say what the effect size is here but the sentence looks unfinished.

Minor

- It wasn't clear to me (maybe I missed it?) why the authors split the manuscripts into those with low / medium / high initial statistical content. I'm not against it, but it comes as a surprise when those results are presented, since that distinction wasn't motivated in advance.
- Pg 9: The authors focus on statistical content within the text only (outside of tables, formulas) and motivate that by wanting to make the analyses comparable across journals, although they also mention wanting to avoid manuscript-specific features. I didn't really understand that justification. For journals, one could simply use journal-specific random effects, which the authors use anyway later on. And content within formulas OR text would be a manuscript-specific feature, so that argument I also found confusing. Also, how was the exclusion of tables and formulas done? – was the software able to do it or was it heuristic somehow?

Appendix E

RSOS-210681.R2 "Does peer review improve the statistical content of manuscripts? A study on 27,467 manuscripts"

Dear Editor,

Thank you for the opportunity to revise and resubmit our manuscript. We appreciate the effort made by reviewers to help us improving our work.

Best regards

Flaminio Squazzoni

Reviewer: 3

Comments to the Author(s)

Attached file: "royal society open.pdf"

This is a very interesting research with rarely accessible data. This paper focused on changes of statistical terms over the process of peer review.. With minor revisions, this paper should be published.

Thank you for encouraging us.

Here are a few points:

Please rephrase the last paragraph of Introduction to focus on what your research is about. This paragraph begins with "Claiming that any could be questionable" and reads confusing.

It will be helpful to state clearly your research questions.

[I do not have access to the revision 2 and associated responses. In my record, I did suggest to clearly state research questions and hypotheses in my first review report.]

We have re-structured Sect 1 to increase the clarity of the text and outline our research questions. We also have revised the sentence.

After accessing to the deposited data, I believe that the results should be definitely boiled down. The total number of records = 11,050 for the four journals measured by six variables:

Independent Variable: Number of reviewers

Independent Variable: Number of rounds of reviews

Independent Variable: Number of statistical Terms (initial submission)

Independent Variable: Max Change of statistical Terms
Independent Variable: Number of statistical Terms (final)
Dependent Variable: Manuscript decision (Accept vs. Reject)
Grouping variable: Journal

Of course, we would love to have a larger and more representative journal sample. However, note that for the type of research here presented, the sample size is remarkable as does its quality: the manuscript files (including rejected manuscripts) are never shared by journals.

Page 4 is very hard to read with all the tracking and colored texts.

We provided a cleaned version of the manuscript.

Page 5 presented model but I could not relate your variables with the model to understand how the results are derived. Please make "observing D" (page 10, line 50) concrete as which above variable is D; also, please be concrete on what variables in "After selecting these variables (page 6, line 19). I feel that these two pages read like teaching the formulas rather than how they are used in this study. It is important to show the use of the formulas with the variables and collected data.

We have simplified the text in the "Statistical models" Section to highlight the variables that were selected. The text is now less academic and focuses more on the two models considered in the paper.

For results why report manuscripts that were not peer reviewed? This is not relevant to your results also not in your analytical dataset. As I quote here "Figure 1 shows that initial submissions had a relatively homogeneous statistical content, except for manuscripts directly accepted by editors without any peer review (see the red solid line, which corresponded to 42 manuscripts)"?
Should be deleted.

We honestly disagree with the reviewer here. We believe that descriptive statistics can be of interest to certain readers and they prepare the background for more robust statistical modeling. Here, we have updated the shared dataset to allow any interested reader to fully reproduce Table 1 and Figure 1 with the complete 27,467 manuscripts.

In Results, you stated "We then considered all 11,243 manuscripts" (page 11, line 56). and later on you explained why it was only 11,050. Let's make two points: first, the

change from 11,243  11,050 be mentioned only once in Data section not in Results; second, it distracts or confuses readers at the time we should focus on the valid dataset.

The restriction to 11,050 manuscripts only applies to the statistical modeling (which discards those missing values) but the rest of the descriptive results refer to all 11,243 manuscripts. We reckon that a step-by-step description of the dataset construction process may be verbose but it also warrants transparency and robustness of the study, as it fully reports how data were extracted from the Royal Society and treated and explain how we did our best to avoid selection bias.

The entire manuscript needs to boil down to minimize repetitive contents.

We have revised the text to avoid any repetition and toned down our claims.

Hope your next revision is for readers rather than for reporting.

Thanks for encouraging us.

Reviewer: 6

Comments to the Author(s)

Attached file: "RSOS210681.JPG"

See Attached

Content of "royal society open.pdf"

I think this is a nice phrase for what to call the measure and it feels appropriately close to the data. However, if no additional analyses/measures are provided, then the framing of this measure should be kept very neutral, i.e. the authors should speak of statistical content increasing or decreasing, but not "improving." I think to frame the changes in content as "improvement" would require more direct measures. Perhaps something like <https://en.wikipedia.org/wiki/Statcheck> or similar tools, if there are any, can be used here? So in my view, the authors face a choice between 1) finding more direct measures of "improvement" (maybe even on a small scale) and then showing that the amount of statistical content correlates with the quality of statistical content or 2) reframing the findings as about peer review increasing the amount of statistical content and speculating, with substantial hedging. At the very least "improve" should be removed from the title, and, if authors insist on having it in

the abstract, it should be something like “peer review increases content...which may signal improvement.”

Thanks for these suggestions. Direct measurements are difficult and in any case suffer from similar limitations of our study. As we suggested in the conclusion, only a sample of human experts who could assess a sample of manuscripts and reports and inform further machine learning analysis could significantly improve the quality of these measurements. We opted for revising the text to avoid any confusion regarding the fact that we did not measure improvements but changes.

I agree with Reviewer 5 that the current justification for the measure as a proxy for statistical quality in the last paragraph before “2. Methods” is insufficient. In fact, I would probably remove that paragraph altogether. I would consider replacing it with reasons we might think that amount of statistical content equal quality of content. For example, I suspect there is literature showing that much statistical analysis is not reported in sufficient detail, so that’s at least one case where more content is widely believed to make manuscripts better. There are also lots of discussions that peer reviewers “force” authors to conduct (unnecessary?) additional experiments and robustness tests. So that’s another way for statistical content to increase in an (arguably) beneficial way.

We tried to improve the Methods section. Being our study a quantitative analysis of explorative measurements (note that this type of research is really at its infancy and so explorative by definition), removing entirely a Methods Section would be problematic. The fact that peer review could force authors to perform irrelevant analysis and statistical tests is probably sometimes true, but it’s something we honestly cannot measure.

Medium issues

- There is a substantial amount of causal language, particularly using the word “effects” to discuss associations, that should be phrased as associations. For example, on page 11, the authors say “The availability of guidelines ... did not have any qualitative effect on the variation...” but ought to say something like “The availability of guidelines ... was not associated with....”

We have revised the text to remove any causal flavor.

- I think the authors should give more weight in the Discussion and maybe Introduction to the possibility that amount of statistical content is a bad

measure of quality, and/or peer review does not improve quality all that much. There are a few pieces of data that point in that direction:

We have added various sentences in the conclusions to discuss this type of study's limitations more extensively.

- Figure 5: In a substantial number of cases, papers that were accepted decreased in statistical content. If more content is better, why would it decrease so often? Perhaps amount of content is a proxy for number of analyses, and reducing these helps focus the paper on a key message.

Note that actually only 25% of accepted manuscripts decreased in statistical content, as we highlighted in the results section when discussing Figure 5a.

- Figure 5: Among rejected papers, there was almost no change in statistical content. If peer review improves content, why would that only appear for accepted papers? If the answer is that the *published* version is necessary for these effects to be visible, then could it be due to copy-editing step rather than peer review?

This is hard to determine. We believe that copy-editing had no effect on this. Note that manuscripts were mostly rejected after the first review round and only rarely passed to further stages. This would explain this. We have outlined this point.

- Overall, the changes in amount of terms seem quite small, so that is worth commenting upon. It seems there is an error in the first paragraph on page 13, "Figure 3 shows...". I assume the authors want to say what the effect size is here but the sentence looks unfinished

We have revised the text here.

Minor

- It wasn't clear to me (maybe I missed it?) why the authors split the manuscripts into those with low / medium / high initial statistical content. I'm not against it, but it comes as a surprise when those results are presented, since that distinction wasn't motivated in advance.

This was to fix certain variables to improve the readability of our 2D plot. We have revised the text

- Pg 9: The authors focus on statistical content within the text only (outside of tables, formulas) and motivate that by wanting to make the analyses

comparable across journals, although they also mention wanting to avoid manuscript-specific features. I didn't really understand that justification. For journals, one could simply use journal-specific random effects, which the authors use anyway later on. And content within formulas OR text would be a manuscript-specific feature, so that argument I also found confusing. Also, how was the exclusion of tables and formulas done? – was the software able to do it or was it heuristic somehow?

We checked for the presence of statistical content within the plain text of the manuscript, also including captions. We skipped equations, figures and tables (sometimes binary or submitted separately) as they differ due to journal and manuscript specific features, such as the file format. Still the statistical information about these elements was gathered as they were referenced and explained within the text and captions. We have changed this paragraph to clarify these points.

Appendix F

RSOS-210681.R2 "Does peer review improve the statistical content of manuscripts? A study on 27,467 manuscripts"

Milan, 23 August 2022

Dear Editor,

Thank you for the opportunity to revise and resubmit our manuscript. We have considered each point from the remaining reviewer carefully. We have revised the text of the abstract and added a track changed version of the manuscript. While we agreed on removing the first sentence and on some linguistic improvements, we decided not to alter this following sentence: "We found that manuscripts with both initial low or high levels of statistical content increased their statistical content during peer review.". This is a clear sentence that reflects our findings while: (1) specifics on the measures we built to capture the statistical content are reported in the paper, not in the abstract; (2) this already said that manuscripts with medium level of initial statistical content did not increase it during peer review (so, it would be superfluous and complicate the flow of the text specifying this in the abstract) . We hope that this acceptable.

Best regards

Flaminio Squazzoni